# Unveiling unique clinical phenotypes of hip fracture patients and the temporal association with cardiovascular events

Warrington W. Q. Hsu [1,6], Xiaowen Zhang [1,6], Chor-Wing Sing[1,2], Kathryn C. B. Tan [3], Ian Chi-Kei Wong [1,2,4,5], Wallis C. Y. Lau [1,2,4,5] & Ching-Lung Cheung [1,2] ✉

Cardiovascular events are the leading cause of death among hip fracture patients. This study aims to identify subphenotypes of hip fracture patients and investigate their association with incident cardiovascular events, all-cause mortality, and health service utilisation in Hong Kong and the United Kingdom populations. By the latent class analysis, we show three distinct clusters in the Hong Kong cohort (n = 78,417): Cluster 1 has cerebrovascular and hypertensive diseases, hyperlipidemia, and diabetes; Cluster 2 has congestive heart failure; Cluster 3 consists of relatively healthy patients. Compared to Cluster 3, higher risks of major adverse cardiovascular events are observed in Cluster 1 (hazard ratio 1.97, 95% CI 1.83 to 2.12) and Cluster 2 (hazard ratio 4.06, 95% CI 3.78 to 4.35). Clusters 1 and 2 are also associated with a higher risk of mortality, more unplanned accident and emergency visits and longer hospital stays. Self-controlled case series analysis shows a significantly elevated risk of major adverse cardiovascular events within 60 days post-hip fracture. Similar associations are observed in the United Kingdom cohort (n = 27,948). Pre-existing heart failure is identified as a unique subphenotype associated with poor prognosis after hip fractures.

Cardiovascular events (CVEs) are the leading cause of death in both the general population and disease populations[1–3]. Temporal association with an increased risk of CVEs has been observed in different disease populations, such as gout[4], influenza-like illness[5] and hip fracture[6]. The incidence of hip fracture, a significant public health concern, was projected to double by 2050 compared to 2018[7]. A meta-analysis of observational studies[8] further confirmed the association between hip fracture and an increased risk of CVEs. Thus, there is an urgent need to enhance the current management of hip fracture patients and improve the prognosis of CVEs following hip fracture.

In our recent study, we demonstrated that there was an immediate increased risk of CVEs after hip fracture[3,6], even after accounting for the post-operative cardiac complications. However, the mechanisms underlying such temporal association are complex and remain unstudied. Considering the diverse clinical characteristics of hip fracture patients, which are crucial for their prognosis, the process of subphenotyping the heterogeneous hip fracture spectrum could provide valuable insights into the temporal association with CVEs and management of patients with hip fracture and CVEs.

Clustering algorithms are commonly used to classify patients with similar characteristics or features. Among various clustering

[1]Department of Pharmacology and Pharmacy, Li Ka Shing Faculty of Medicine, The University of Hong Kong, Hong Kong Special Administrative Region, China. [2]Laboratory of Data Discovery for Health Limited (D24H), Hong Kong Science Park, Hong Kong Special Administrative Region, China. [3]Department of Medicine, Li Ka Shing Faculty of Medicine, The University of Hong Kong, Hong Kong Special Administrative Region, China. [4]Research Department of Practice and Policy, School of Pharmacy, University College London, London, UK. [5]Centre for Medicines Optimisation Research and Education, University College London Hospitals NHS Foundation Trust, London, UK. [6]These authors contributed equally: Warrington W. Q. Hsu, Xiaowen Zhang. ✉e-mail: lung1212@hku.hk

algorithms, latent class analysis (LCA), an unsupervised machine learning technique for classifying subjects into clusters by a combination of variables[9–11], has been shown to be the optimum clustering algorithm for health records[12]. Therefore, in this study, we first utilised the unsupervised LCA to subgroup hip fracture patients who share common clinical characteristics in two extensive hip fracture cohorts: one from Hong Kong (HK; $N = 78,417$) and the other from the United Kingdom (UK; $N = 27,948$). Subsequently, to uncover variations in long-term outcomes for hip fracture patients based on their subphenotypic classification, conventional survival analyses (between-individual comparison) and self-controlled case series (SCCS; within-individual comparison) were conducted. The analyses aimed to quantify the risk of CVE-related outcomes, all-cause mortality, and health service utilisation following hip fracture and to evaluate potential differences across the identified subphenotypes. Unlike the conventional survival analyses comparing the prognosis risk between individuals, SCCS analyses focused only on hip fracture patients who experienced the prognostic events with each patient serving as their own control, which effectively controlled for time-fixed confounders and between-individual differences[13]. Consequently, SCCS analyses enabled the detection of temporal patterns within each LCA-derived cluster and facilitated the identification of potential differences in temporal patterns across the clusters, which cannot be captured by conventional survival analyses alone. Overall, the conventional survival analysis and SCCS are complementary approaches that provide a comprehensive understanding of the prognosis risk and temporal associations in the identified subphenotypes.

Here, we show findings on subphenotypes of hip fracture patients in both the Hong Kong and the UK older adult populations using LCA. Heart failure consistently emerges as a key characteristic associated with poor prognosis in hip fracture patients. Temporal associations with CVEs in all hip fracture patient subphenotypes are also observed.

## Results

### Identification of subphenotypes

The LCA was conducted to identify hip fracture subphenotypes, followed by descriptive analysis to investigate the baseline characteristics of the identified subphenotypes. The Hong Kong Clinical Data Analysis and Reporting System (HK CDARS) hip fracture cohort consisted of 78,417 patients (54,892 and 23,525 patients for training and internal test sets, respectively) with hip fracture aged ≥65 years (Supplementary Fig. 1a). The mean age for the entire CDARS cohort was 83.43 years (SD = 7.52) and 68.6% of the cohort were female (Supplementary Table 1). The United Kingdom Health Improvement Network (UK THIN) cohort included 27,948 hip fracture patients aged ≥65 years (Supplementary Fig. 1b), with a mean age of 82.48 years (SD = 7.67) and 74.3% of the cohort being females. All Spearman's rank correlation coefficients among the clustering variables were <0.5, therefore no clustering variables were removed.

The statistics of model fitting performance across the number of clusters are shown in Supplementary Fig. 2. The optimal number of clusters was selected to be three in both the HK CDARS training and test sets (Supplementary Fig. 2a, b) and two in the UK THIN cohort (Supplementary Fig. 2c). To assess the uncertainty of cluster membership, the distribution of each subject's highest class membership probability (the likelihood that reflects a subject's most likely cluster assignment) was evaluated (Supplementary Table 2). This evaluation was conducted by computing the median, and the lower and upper quartiles of these probabilities for all subjects within each cluster. The medians of the highest class membership probabilities ranged from 0.82 to 0.98 across clusters. These median values indicated a moderately high to very high degree of certainty in the assignment of subjects to their respective clusters.

The baseline characteristics of the three LCA-generated clusters in the HK CDARS training set resembled that of the three clusters

identified in the CDARS test set (Supplementary Fig. 3a, b), with Pearson's correlation coefficients of 1.00 between the corresponding clusters in training and test sets (Supplementary Table 3). This supported the decision to merge the two datasets in the subsequent prognosis analyses. Cluster 1 ($n = 11,886$ [21.65%] for training set; $n = 4876$ [20.73%] for test set) consisted of a high prevalence of cerebrovascular diseases, hypertensive diseases, hyperlipidemia, and diabetes (standardised mean differences [SMD] > 0.8 vs Cluster 3; Tables 1 and 2). Cluster 2 ($n = 6588$ [12.00%] for training set; $n = 3272$ [13.91%] for test set) consisted of a high prevalence of coronary heart disease, congestive heart failure, hypertensive diseases, and arrhythmia and conduction disorders (SMD > 0.8 vs Cluster 3; Tables 1 and 2). Cluster 3 ($n = 36,418$ [66.34%] for the training set; $n = 15,377$ [65.36%] for the test set) was relatively healthy with the lowest prevalence of most clinical conditions studied. Notably, congestive heart failure was almost exclusively present in Cluster 2 but not in other clusters.

In the UK THIN dataset, Cluster 1 consisted of a high prevalence of CVE including coronary heart disease, congestive heart failure, and arrhythmia and conduction disorders (SMD > 0.8), and with a higher prevalence of most clinical conditions studied when compared with the relatively healthy Cluster 2 (Table 3 and Supplementary Fig. 3c). Consistently, congestive heart failure was exclusively present in Cluster 1. Using SMD > 0.8 as the criterion to identify the characteristic variables of large differences between clusters, the baseline characteristics of Cluster 1 in UK THIN (variables with SMD > 0.8 in the SMD column in Table 3) resembled that of Cluster 2 in HK CDARS (variables with SMD > 0.8 in the SMD Cluster 2 vs 3 columns in Tables 1 and 2), except for hypertensive diseases. The two clusters were highly correlated with a Pearson's correlation coefficient of 0.71. Cluster 2 in UK THIN and Cluster 3 in HK CDARS (the relatively healthy clusters) were also strongly correlated, with a Pearson's correlation coefficient of 0.93.

### Prognosis of subphenotypes

The associations between the hip fracture subphenotypes and prognostic outcomes of interest were investigated using conventional survival analysis, with the relatively healthy cluster as the reference groups in both HK CDARS (Cluster 3) and UK THIN (Cluster 2) cohorts (Table 4 and Fig. 1). In the HK CDARS, the risk of 180-day all-cause mortality and major adverse cardiovascular events (MACE) were significantly higher in Cluster 1 (all-cause mortality: hazard ratio [HR] 1.35, 95% CI 1.28 to 1.42; MACE: HR 1.97, 95% CI 1.83 to 2.12) and Cluster 2 (all-cause mortality: HR 2.22, 95% CI 2.10 to 2.34; MACE: HR 4.06, 95% CI 3.78 to 4.35) compared to Cluster 3. For the secondary outcomes, both Cluster 1 and Cluster 2 were associated with a higher number of hospital visits, accident and emergency (A&E) visits, and total length of hospital stays in the 180-day period after hip fracture (Table 4 and Fig. 1b).

For the UK THIN cohort, Cluster 1 was associated with an increased risk of 180-day all-cause mortality (HR 1.38, 95% CI 1.28 to 1.49) and MACE (HR 1.84, 95% CI 1.53 to 2.23) when compared to the relatively healthy cluster (Table 4 and Fig. 1a). In both the HK CADRS and UK THIN cohorts, similar results were observed for the individual MACE outcomes (Supplementary Table 4) and in the sensitivity analysis (Supplementary Tables 5–7). The increased risk of MACE and all-cause mortality reported in the main analysis in Cluster 1 and 2 in HK CDARS, and Cluster 1 in UK THIN, were in general consistently observed across all the age, sex and surgery type subgroups (Supplementary Table 8). However, differences in the magnitude of these risks were observed when comparing the different subgroups. The stratified analysis generally showed that females, younger patients, and those undergoing partial hip replacement surgeries tended to exhibit higher risks, as reflected by HRs, when compared with the male, older, and internal fixation counterparts. In particular, in Cluster 2 in HK CDARS and Cluster 1 in UK THIN, the HR for the association between the hip

**Table 1 | Baseline characteristics of hip fracture subphenotypes (HK CDARS training set)**

| Variables | HK CDARS training set | | | | | |
|---|---|---|---|---|---|---|
| | Cluster 1 | Cluster 2 | Cluster 3 | SMD Cluster 1 vs 3 (p-value[a]) | SMD Cluster 2 vs 3 (p-value[a]) | SMD Cluster 1 vs 2 (p-value[a]) |
| n | 11,886 | 6588 | 36,418 | | | |
| Males, n (%) | 3944 (33.2) | 2251 (34.2) | 11,108 (30.5) | 0.058 (<0.001) | 0.078 (<0.001) | 0.021 (0.179) |
| Age, mean (SD) | 82.38 (7.21) | 85.74 (6.84) | 83.32 (7.67) | 0.126 (<0.001) | 0.333 (<0.001) | 0.478 (<0.001) |
| **Diagnosis record within 5 years before index date, n (%)** | | | | | | |
| Coronary heart disease | 2018 (17.0) | 3195 (48.5) | 914 (2.5) | 0.503 (<0.001) | 1.242 (<0.001) | 0.713 (<0.001) |
| Congestive heart failure | 5 (0.0) | 4954 (75.2) | 386 (1.1) | 0.138 (<0.001) | 2.362 (<0.001) | 2.458 (<0.001) |
| Cerebrovascular diseases | 3885 (32.7) | 976 (14.8) | 1231 (3.4) | 0.824 (<0.001) | 0.406 (<0.001) | 0.430 (<0.001) |
| Hypertensive diseases | 10629 (89.4) | 4680 (71.0) | 5286 (14.5) | 2.266 (<0.001) | 1.392 (<0.001) | 0.474 (<0.001) |
| Arrhythmia and conduction disorders | 1338 (11.3) | 3561 (54.1) | 1732 (4.8) | 0.241 (<0.001) | 1.287 (<0.001) | 1.026 (<0.001) |
| Arterial disease | 1398 (11.8) | 463 (7.0) | 172 (0.5) | 0.485 (<0.001) | 0.350 (<0.001) | 0.163 (<0.001) |
| Chronic obstructive pulmonary disease | 852 (7.2) | 1721 (26.1) | 2451 (6.7) | 0.017 (0.105) | 0.542 (<0.001) | 0.526 (<0.001) |
| Hyperlipidemia | 3706 (31.2) | 1067 (16.2) | 0 (0.0) | 0.952 (<0.001) | 0.622 (<0.001) | 0.358 (<0.001) |
| Obesity | 186 (1.6) | 60 (0.9) | 0 (0.0) | 0.178 (<0.001) | 0.136 (<0.001) | 0.059 (<0.001) |
| Diabetes | 6680 (56.2) | 2204 (33.5) | 1807 (5.0) | 1.338 (<0.001) | 0.776 (<0.001) | 0.470 (<0.001) |
| Thyroid disorders | 350 (2.9) | 413 (6.3) | 243 (0.7) | 0.172 (<0.001) | 0.310 (<0.001) | 0.159 (<0.001) |
| Chronic renal disease | 1420 (11.9) | 1395 (21.2) | 172 (0.5) | 0.490 (<0.001) | 0.707 (<0.001) | 0.250 (<0.001) |
| Liver diseases | 137 (1.2) | 84 (1.3) | 141 (0.4) | 0.088 (<0.001) | 0.098 (<0.001) | 0.011 (0.508) |
| Osteoporosis | 568 (4.8) | 421 (6.4) | 1149 (3.2) | 0.083 (<0.001) | 0.152 (<0.001) | 0.070 (<0.001) |
| Paget's disease of bone | 6 (0.1) | 6 (0.1) | 2 (0.0) | 0.027 (0.004) | 0.039 (<0.001) | 0.015 (0.462) |
| Major fractures other than hip fracture | 1011 (8.5) | 607 (9.2) | 2845 (7.8) | 0.025 (0.016) | 0.050 (<0.001) | 0.025 (0.109) |
| Connective tissue disease | 82 (0.7) | 67 (1.0) | 233 (0.6) | 0.006 (0.601) | 0.042 (0.001) | 0.036 (0.022) |
| Osteoarthritis | 1280 (10.8) | 870 (13.2) | 1816 (5.0) | 0.216 (<0.001) | 0.289 (<0.001) | 0.075 (<0.001) |
| Depression | 388 (3.3) | 137 (2.1) | 426 (1.2) | 0.143 (<0.001) | 0.072 (<0.001) | 0.074 (<0.001) |
| Dementia | 1320 (11.1) | 563 (8.5) | 1982 (5.4) | 0.207 (<0.001) | 0.122 (<0.001) | 0.086 (<0.001) |

n number of independent patients, % percentage, SD standard deviation, SMD standardized mean difference.
[a]P-values computed from group comparison tests (Chi-square tests for categorical variables and one-way ANOVA for continuous variables).

fracture subphenotype and MACE was observed to be higher in females when compared to males (interaction p-value < 0.05).

**Results from the temporal analysis**

The competing risk regression and SCCS were used to investigate the temporal associations between the hip fracture subphenotypes and MACE. In the between-individual analysis using competing risk regression, an immediate risk of overall MACE was observed in the HK CDARS and UK THIN cohorts using the relatively healthy cluster as the reference (Supplementary Fig. 4). The results of the temporal association with individual MACE outcomes are provided in Supplementary Table 9.

In the within-individual comparison using the SCCS analysis, we included 6366 patients with MACE that occurred during the observation periods in the HK CDARS (Table 5) over a total follow-up time of 19154.64 person-year. Similar temporal association patterns were observed in all clusters, including the relatively healthy cluster. Specifically, the age-adjusted incidence rate ratios (IRRs) of MACE were found to be the highest and statistically significant at 1–60 days after hip fracture across the three clusters (Cluster 1: IRR 1.84, 95% CI 1.54 to 2.20; Cluster 2: IRR 1.99, 95% CI 1.62 to 2.45; Cluster 3: IRR 2.05, 95% CI 1.81 to 2.33), and the IRRs decreased during the subsequent post-hip fracture risk periods (Fig. 2a and Table 5). The cluster-specific patterns matched the overall temporal pattern of the entire cohort, with the age-adjusted IRR of MACE at 1–60 days after hip fracture being 1.93 (95% CI, 1.76 to 2.11; Fig. 2a and Table 5). Similar results were observed in the UK THIN database, where significantly higher IRRs of MACE were observed at 1–60 days after hip fracture in the two clusters (Cluster 1:

IRR 3.35, 95% CI 2.28 to 4.92; Cluster 2: IRR 2.25, 95% CI 1.81 to 2.81), matching the overall pattern of the entire cohort (IRR 2.43, 95% CI 2.01 to 2.94; Fig. 2b and Table 6). In both the HK CDARS and UK THIN within-individual analyses, similar conclusions were observed with the individual MACE outcomes, and the sensitivity analyses with shorter exposure intervals and considering only the post-hip fracture period as the baseline (Supplementary Table 10 and Supplementary Fig. 5). Notably, the higher risk of MACE was observed in the intervals of 1–30 d and 31–60 d in both the HK CDARS and UK THIN cohorts. In addition, we plotted the incidence rates of MACE within one year following hip fracture (Supplementary Fig. 6). The plots showed similar temporal patterns across clusters, but there were notable differences in the incidence rates across clusters in both HK CDARS and UK THIN, which aligned with the results from competing risk regression and SCCS.

Furthermore, to verify the hip fracture subphenotypes were not direct replications of conventional cardiovascular disease (CVD) groups, a comparative analysis was conducted using a reference CDARS cohort composed of patients with myocardial infarction (MI). In this analysis, the 60-day and 12-month event rates for MACE and all-cause mortality for the subphenotypes were compared to that of the reference MI cohort (Supplementary Table 11). The MI cohort revealed expectedly high event rates of MACE and all-cause mortality. Notably, the MI cohort showed a moderate increase (28.48%) in the event rates of MACE (18.89%–24.27%) and a substantial rise (137.8%) in mortality event rates (11.77%–27.99%) from the 60-day to the 1-year period. Conversely, the hip fracture cohort, with 3.18% of patients with baseline MI, demonstrated a sharper increase in both outcomes over the same period, with the MACE event rates more than doubling (164.2%

**Table 2 | Baseline characteristics of hip fracture subphenotypes (HK CDARS test set)**

| Variables | HK CDARS test set | | | | | |
| --- | --- | --- | --- | --- | --- | --- |
| | Cluster 1 | Cluster 2 | Cluster 3 | SMD Cluster 1 vs 3 (*p*-value[a]) | SMD Cluster 2 vs 3 (*p*-value[a]) | SMD Cluster 1 vs 2 (*p*-value[a]) |
| n | 4876 | 3272 | 15,377 | | | |
| Males, n (%) | 1517 (31.1) | 1082 (33.1) | 4692 (30.5) | 0.013 (0.440) | 0.055 (0.004) | 0.042 (0.067) |
| Age, mean (SD) | 82.23 (7.20) | 86.27 (6.63) | 83.28 (7.62) | 0.141 (<0.001) | 0.419 (<0.001) | 0.584 (<0.001) |
| **Diagnosis record within 5 years before index date, n (%)** | | | | | | |
| Coronary heart disease | 624 (12.8) | 1616 (49.4) | 341 (2.2) | 0.410 (<0.001) | 1.280 (<0.001) | 0.861 (<0.001) |
| Congestive heart failure | 0 (0.0) | 2074 (63.4) | 186 (1.2) | 0.156 (<0.001) | 1.780 (<0.001) | 1.861 (<0.001) |
| Cerebrovascular diseases | 1654 (33.9) | 475 (14.5) | 502 (3.3) | 0.857 (<0.001) | 0.403 (<0.001) | 0.465 (<0.001) |
| Hypertensive diseases | 4303 (88.2) | 2364 (72.2) | 2144 (13.9) | 2.222 (<0.001) | 1.457 (<0.001) | 0.410 (<0.001) |
| Arrhythmia and conduction disorders | 466 (9.6) | 1724 (52.7) | 541 (3.5) | 0.246 (<0.001) | 1.307 (<0.001) | 1.053 (<0.001) |
| Arterial disease | 618 (12.7) | 197 (6.0) | 77 (0.5) | 0.506 (<0.001) | 0.315 (<0.001) | 0.230 (<0.001) |
| Chronic obstructive pulmonary disease | 269 (5.5) | 790 (24.1) | 1043 (6.8) | 0.053 (0.002) | 0.495 (<0.001) | 0.543 (<0.001) |
| Hyperlipidemia | 1575 (32.3) | 545 (16.7) | 0 (0.0) | 0.977 (<0.001) | 0.632 (<0.001) | 0.370 (<0.001) |
| Obesity | 83 (1.7) | 26 (0.8) | 0 (0.0) | 0.186 (<0.001) | 0.127 (<0.001) | 0.082 (0.001) |
| Diabetes | 2800 (57.4) | 991 (30.3) | 756 (4.9) | 1.376 (<0.001) | 0.707 (<0.001) | 0.569 (<0.001) |
| Thyroid disorders | 162 (3.3) | 187 (5.7) | 102 (0.7) | 0.191 (<0.001) | 0.291 (<0.001) | 0.115 (<0.001) |
| Chronic renal disease | 569 (11.7) | 621 (19.0) | 81 (0.5) | 0.479 (<0.001) | 0.654 (<0.001) | 0.204 (<0.001) |
| Liver diseases | 43 (0.9) | 42 (1.3) | 62 (0.4) | 0.060 (<0.001) | 0.096 (<0.001) | 0.039 (0.101) |
| Osteoporosis | 205 (4.2) | 176 (5.4) | 509 (3.3) | 0.047 (0.004) | 0.102 (<0.001) | 0.055 (0.016) |
| Paget's disease of bone | 3 (0.1) | 0 (0.0) | 1 (0.0) | 0.030 (0.072) | 0.011 (1.000) | 0.035 (0.406) |
| Major fractures other than hip fracture | 453 (9.3) | 310 (9.5) | 1173 (7.6) | 0.060 (<0.001) | 0.066 (<0.001) | 0.006 (0.810) |
| Connective tissue disease | 71 (1.5) | 25 (0.8) | 70 (0.5) | 0.103 (<0.001) | 0.040 (0.034) | 0.066 (0.006) |
| Osteoarthritis | 579 (11.9) | 438 (13.4) | 737 (4.8) | 0.258 (<0.001) | 0.302 (<0.001) | 0.046 (0.047) |
| Depression | 141 (2.9) | 67 (2.0) | 185 (1.2) | 0.119 (<0.001) | 0.067 (<0.001) | 0.054 (0.022) |
| Dementia | 504 (10.3) | 313 (9.6) | 930 (6.0) | 0.157 (<0.001) | 0.131 (<0.001) | 0.026 (0.273) |

*n* number of independent patients, *%* percentage, *SD* standard deviation, *SMD* standardized mean difference.

[a]*P*-values computed from group comparison tests (Chi-square tests for categorical variables and one-way ANOVA for continuous variables).

increase, from 2.29 to 6.05%) and all-cause mortality rates tripling (255.2% increase, from 4.67% to 16.59%). This trend was consistent across all three clusters within the hip fracture cohort. In particular, Cluster 2 showed the highest absolute event rates among the hip fracture clusters, with the 1-year MACE rate being 10.72%, almost half of that of the MI cohort, even though only 15.04% of patients in Cluster 2 had baseline MI. In addition, the 1-year mortality rate in Cluster 2 surpassed that of the MI cohort (31.35% vs 27.99%).

## Discussion

We robustly segmented the hidden clinical characteristics of hip fracture patients into distinct clusters (i.e., subphenotypes) in both the Hong Kong and UK populations using LCA and demonstrated that these clusters explained, at least partially, the observed temporal association between hip fracture subphenotypes and CVEs. Notably, one of the clusters predominant by heart failure was correlated with a poor prognosis characterised by more unplanned A&E visits, prolonged hospitalisation, and higher risk of all-cause mortality and MACE in comparison to the relatively healthy reference group. In addition, a consistent temporal association between the hip fracture subphenotypes and MACE was observed in the Hong Kong and UK populations, even after accounting for the potential post-operative cardiac complications.

This retrospective study implemented the unsupervised machine learning approach of LCA on two population-based electronic health record (EHR) databases in Hong Kong and the UK to identify hip fracture subphenotypes. Clusters with a high prevalence of CVEs were identified in the two independent population cohorts (Cluster 1 and 2

in HK CDARS and Cluster 1 in UK THIN). These clusters were associated with poor health prognoses, with an increased risk of 180-day all-cause mortality and MACE. Notably, Cluster 2 in the HK CDARS and Cluster 1 in the UK THIN were highly similar, in which both clusters had a higher prevalence of coronary heart disease, congestive heart failure and arrhythmia and conduction disorders compared with the reference cluster (SMD > 0.8). Particularly, these clusters were predominated by heart failure. Two meta-analyses reported a significantly increased risk of hip fracture in patients with heart failure[14,15]. Previous studies also showed an association between heart failure and decreased bone mineral density (BMD) at the hip and femoral neck[16,17]. However, the detailed mechanism underlying the exclusiveness of heart failure in the cluster with hip fracture requires further study.

In comparison of the relatively healthy cluster, Cluster 1 and 2 in the Hong Kong cohort (and Cluster 1 in the UK cohort) were associated with an increased risk of 180-day mortality and MACE. Previous population-based cohort studies showed evidence of an association between a history of cardiovascular disease and increased mortality risk in patients with hip fracture[18,19]. A Danish study showed that a history of CVEs and cardiovascular biomarkers were associated with mortality within the 30-day period after hip fracture, with heart failure demonstrating the strongest association with mortality when compared to other CVEs[18]. Another study indicated that heart failure patients with hip fracture experienced an over two-fold increased risk of mortality compared to heart failure patients without hip fracture[20]. The results of previous studies were in line with our LCA results, suggesting that pre-existing CVEs, particularly heart failure, were associated with a poorer post-hip fracture prognosis in both the Hong

**Table 3 | Baseline characteristics of hip fracture sub-phenotypes (UK THIN cohort)**

| Variables | UK THIN cohort | | |
|---|---|---|---|
| | Cluster 1 | Cluster 2 | SMD (p-value[a]) |
| n | 4966 | 22,982 | |
| Males, n (%) | 1910 (38.5) | 5274 (22.9) | 0.341 (<0.001) |
| Age, mean (SD) | 83.77 (6.85) | 82.20 (7.81) | 0.213 (<0.001) |
| **Diagnosis record within 5 years before index date, n (%)** | | | |
| Coronary heart disease | 2147 (43.2) | 1090 (4.7) | 1.010 (<0.001) |
| Congestive heart failure | 1502 (30.2) | 0 (0.0) | 0.931 (<0.001) |
| Cerebrovascular diseases | 1322 (26.6) | 1525 (6.6) | 0.557 (<0.001) |
| Hypertensive diseases | 997 (20.1) | 2658 (11.6) | 0.235 (<0.001) |
| Arrhythmia and conduction disorders | 2263 (45.6) | 1108 (4.8) | 1.063 (<0.001) |
| Arterial disease | 827 (16.7) | 600 (2.6) | 0.490 (<0.001) |
| Chronic obstructive pulmonary disease | 965 (19.4) | 2085 (9.1) | 0.300 (<0.001) |
| Hyperlipidemia | 396 (8.0) | 686 (3.0) | 0.221 (<0.001) |
| Obesity | 141 (2.8) | 202 (0.9) | 0.146 (<0.001) |
| Diabetes | 727 (14.6) | 1125 (4.9) | 0.333 (<0.001) |
| Thyroid disorders | 779 (15.7) | 1866 (8.1) | 0.235 (<0.001) |
| Chronic renal disease | 1977 (39.8) | 3031 (13.2) | 0.633 (<0.001) |
| Liver diseases | 33 (0.7) | 66 (0.3) | 0.055 (<0.001) |
| Osteoporosis | 548 (11.0) | 2669 (11.6) | 0.018 (0.257) |
| Paget's disease of bone | 22 (0.4) | 28 (0.1) | 0.061 (<0.001) |
| Major fractures other than hip fracture | 501 (10.1) | 2503 (10.9) | 0.026 (0.103) |
| Connective tissue disease | 256 (5.2) | 781 (3.4) | 0.087 (<0.001) |
| Osteoarthritis | 1221 (24.6) | 3719 (16.2) | 0.210 (<0.001) |
| Depression | 532 (10.7) | 1324 (5.8) | 0.181 (<0.001) |
| Dementia | 551 (11.1) | 2537 (11.0) | 0.002 (0.928) |

n number of independent patients, % percentage, SD standard deviation, SMD standardized mean difference.

[a]P-values computed from group comparison tests (Chi-square tests for categorical variables and one-way ANOVA for continuous variables).

Kong and UK cohorts. A sex-specific risk of MACE has been observed in Cluster 2 in HK CDARS and Cluster 1 in UK THIN. While the male sex is often known to be associated with a higher risk of post-fracture morbidity including CVE[21], our subgroup analysis showed that in Cluster 2 in HK CDARS and Cluster 1 in UK THIN, the HR for the association between the hip fracture subphenotype and MACE was higher in females than in males. Therefore, it is important to further consider the factor of sex when evaluating the prognosis for those within the already high-risk subphenotypes.

Temporal associations of hip fracture subphenotypes with CVEs were observed. In the between-individual analysis, the risks of MACEs in Clusters 1 and 2 in HK CDARS cohort and Cluster 1 in UK THIN cohort were higher than that of the relatively healthy clusters, and the risks decreased within one year following the hip fracture. Such findings could be driven by the higher prevalence of prevalent cardiac events in these clusters and other potential unmeasured confounders. However, the differences in the event rates of MACE and all-cause mortality between the hip fracture subphenotypes and the reference group of MI patients partially supported that the subphenotypes identified by LCA were not simply reiterations of traditional CVD patient groups. We further conducted SCCS analyses to address the between-person residual confounding and to investigate the temporal patterns within each identified cluster, which cannot be achieved through conventional survival analyses. While the results revealed similar temporal association patterns across clusters, consistently showing the highest

risk of MACE within 1–60 days after the hip fracture, it is important to note that the risk of post-hip fracture MACE remained elevated even among individuals classified as relatively healthy. Furthermore, a higher risk of MACE was also observed in the interval of 31–60 and 61–90 days, which were periods unlikely to be affected by post-operative cardiac complications. Collectively, although the temporal patterns across clusters were similar, the risk of post-hip fracture MACE in the clusters varied. Specifically, in HK CDARS, Cluster 2 exhibited the highest risk, followed by Cluster 1 and the relatively healthy cluster. In UK THIN, Cluster 1 had a higher risk compared to the relatively healthy cluster. These findings suggested that hip fracture per se could lead to short-term elevated risk of MACE and such temporal elevated risk could be exacerbated by the presence of pre-existing cardiac diseases and other comorbidities.

Our study has several strengths. First, this study implemented the approach of LCA to investigate the heterogeneity among hip fracture patients. LCA enables the identification of patient subgroups with similar characteristics based on clinical and demographic variables, without requiring predefined hypotheses or assumptions from researchers. The risk and prognosis of different subphenotypes were then evaluated through statistical methods including competing risk regression and SCCS. Another major strength is the inclusion of the UK THIN cohort, which allowed the validation of the hip fracture subphenotypes identified in the HK CDARS cohort. Medical studies utilising clustering analysis (and machine learning in general) often rely on homogenous patient samples, which limits the generalisability of the identified clusters. This study effectively addressed the concern by incorporating the two diverse population cohorts. Further strengths of this study include the large sample sizes and the utilisation of comprehensive statistical methods. Competing risk regression was used to account for the competing event of death when computing hazard ratios. The design of SCCS inherently controlled for the time-invariant covariates and allowed the temporal association between hip fracture and MACE to be evaluated. The SCCS design outperforms other observational study designs in terms of efficiency and precision in estimating exposure effects[22].

There are limitations to this study. First, this study mainly used predefined baseline diagnosis variables for clustering, and other types of variables such as socioeconomic factors and laboratory features were not available. Additional clustering variables could potentially modify the assignment of hip fracture subphenotypes or reveal the subphenotypes. However, using diagnosis variables as clustering variables offers accessibility and convenience compared to variables not routinely collected across EHR systems from different countries and clinical settings. This facilitates external validation of the identified subphenotypes and enables timely subphenotype assignment in clinical care for patients with hip fracture. Secondly, in terms of SCCS, patients with MACE that occurred on day 0 were excluded due to the uncertainty of whether such a record should be considered a pre- or post-exposure event. This might bias the IRR estimates in the first risk period but was more likely to be an underestimation. In addition, older adults with pre-existing MACE were more likely to experience MACE as a postoperative complication after hip fracture[23,24]. This issue was addressed by including only the first MACE and using shorter time intervals. The first 60-day risk period was separated into 1–30 and 31–60 days. The generally significantly high IRRs in the 31–60 day risk period, not just 1–30 day, suggest the associations were unlikely due to postoperative complications. Third, due to the use of different diagnosis coding systems in the CDARS and THIN databases, the definition of baseline conditions may vary between the two cohorts. Measures were taken to ensure consistency, including referencing the health data research (HDR) UK Phenotype Library for disease definitions and utilising the UK Biobank's resource to translate ICD-9 codes from CDARS to Read codes in THIN. Additionally, the clustering solutions generated by LCA were not identical in the two cohorts. The HK CDARS

**Table 4 | The association between hip fracture subphenotypes and 180-day outcomes of interest**

| | HK CDARS[a] | | | | UK THIN[b] | |
|---|---|---|---|---|---|---|
| | Cluster 1 vs 3 (ref) | | Cluster 2 vs 3 (ref) | | Cluster 1 vs 2 (ref) | |
| **Clinical outcomes** | HR (95% CI) | *p*-value | HR (95% CI) | *p*-value | HR (95% CI) | *p*-value |
| All-cause mortality[c] | 1.35 (1.28–1.42) | <0.001 | 2.22 (2.10–2.34) | <0.001 | 1.38 (1.28–1.49) | <0.001 |
| MACE[d] | 1.97 (1.83–2.12) | <0.001 | 4.06 (3.78–4.35) | <0.001 | 1.84 (1.53–2.23) | <0.001 |
| **Hospital outcomes[e]** | IRR (95% CI) | *p*-value | IRR (95% CI) | *p*-value | IRR (95% CI) | *p*-value |
| Number of hospital visits | 1.49 (1.47–1.51) | <0.001 | 2.02 (1.99–2.06) | <0.001 | NA | NA |
| Number of A&E visits | 1.47 (1.44–1.50) | <0.001 | 2.04 (1.99–2.09) | <0.001 | NA | NA |
| Total length of hospital stays in days | 1.32 (1.32–1.33) | <0.001 | 1.74 (1.73–1.75) | <0.001 | NA | NA |

*HR* hazard ratio, *IRR* incidence rate ratio, *MACE* major adverse cardiovascular events, *A&E* accident and emergency, *CI* confidence interval.
[a]*n* = 78,417 independent patients in HK CDARS (*n* = 16,762 in Cluster 1; *n* = 9860 in Cluster 2; *n* = 51,795 in Cluster 3).
[b]*n* = 27,948 independent patients in UK THIN (*n* = 4966 in Cluster 1; *n* = 22,982 in Cluster 2).
[c]HRs, associated 95% CIs and two-sided *p*-values are derived from Cox proportional regression, with adjustment of age and sex.
[d]HRs, associated 95% CIs and two-sided *p*-values are derived from competing risk regression, with adjustment of age and sex.
[e]IRRs, associated 95% CIs and two-sided *p*-values are derived from Poisson regression, with adjustment of age and sex.

and UK THIN databases differed significantly not only in terms of clinical settings (hospital-based in HK CDARS versus primary care in UK THIN), but also in the demographic composition of their populations (predominantly Asian Chinese in HK CDARS versus predominantly Caucasian in UK THIN), data entry behaviours, and diagnosis coding systems. These diverse factors inherent to EHR databases could affect the generalisability of LCA models and survival analysis results. These variations may also explain the presence of an additional CVE cluster in the Hong Kong cohort, potentially attributed to a more comprehensive and timely capture of severe conditions in the hospital-based CDARS when compared to the primary care-based THIN. Moreover, underreporting of diagnoses due to factors such as underdiagnosis and undercoding is a common issue in EHR databases. The absence of a diagnosis code in the EHR databases, such as in undiagnosed patients, would result in the condition being classified as not present under our study's definition of diagnosis variables. For example, our reported baseline hypertensive disease prevalence of 13.1% in the UK THIN cohort was much lower than the prevalence of 29.7% in the general UK adult population reported by Health Survey for England (HSE)[25]. This underestimation aligned with a study validating hypertension diagnosis coding in the UK THIN database, which reported an underestimated prevalence of 14.0% using Read codes to define hypertension[25]. The differences in the prevalence of clustering variables (such as hypertension) between HK CDARS and UK THIN could potentially contribute to the differences in the resulting clustering solutions. Therefore, a cautious interpretation of the clustering solutions is required. However, despite the variations between HK CDARS and UK THIN, our findings demonstrated consistency in several key aspects across the two cohorts, including the identification of subphenotypes with a pronounced presence of baseline CVE, and a temporal association between hip fracture subphenotypes and MACE. To reinforce the robustness of the LCA models, further independent validations across EHR databases and populations are recommended, as consistent clustering results in diverse settings would support the generalisability of the LCA models. Future studies could also explore alternative definitions of conditions using data beyond diagnosis records.

This study has important implications. Notably, the LCA results revealed subphenotypes with pronounced baseline CVE profiles across two independent population-based cohorts, without any preconceived assumption about CVE history being particularly important among the clustering variables. Although current clinical guidelines highlight the importance of perioperative evaluation and optimisation for patients with a high cardiac risk[26,27], there is a noticeable lack of emphasis on CVE risk management in the specific context of hip fracture care. For example, current heart failure

management guidelines rarely cover the management of osteoporosis and hip fracture risk in heart failure patients[28,29]. Our findings supported by the previous studies emphasise the importance of taking measures to mitigate the risk of hip fracture among heart failure patients, given the potential poor prognosis after hip fracture. Notably, our temporal analysis showed an immediate increased risk of MACE after hip fracture, with the incidence rates of Cluster 2 in HK CDARS and Cluster 1 in UK THIN being particularly high. These findings underscore the importance of prompt CVE risk management especially in these high-risk patient groups.

The elevated CVE risk after hip fracture is inadequately documented in the literature, along with the association between pre-existing CVE and the subsequent risk of CVEs in hip fracture patients, especially considering multimorbidity is common in the older hip fracture patient population[30]. Our study addressed this gap by quantifying the associated risks, and the elevated risk of MACE within 60 days post-hip fracture across all subphenotypes highlighted again the critical need for integrating CVE management into the broader hip fracture care strategy within the first year after hip fracture. Using the relatively healthy cluster as the reference group, this comparative framework allowed us to evaluate the relative prognosis of each unique comorbidity cluster, thereby laying the groundwork for more personalised management strategies and prioritisation of healthcare resources.

Hip fractures are often managed as a homogenous condition, yet the current study explored the heterogeneity among hip fracture patients and reported variations in mortality, CVE risk and hospital utilisation outcomes across subphenotypes. This variation emphasises the need to advance from conventional stratifications, such as those relying on a single baseline condition or demographic factor, to a more holistic, multimorbidity-focused approach in classifying patients, as facilitated by LCA. To operationalise the subphenotyping process, a digital tool could be developed and incorporated into the electronic clinical management system for personalised management of the hip fracture patients based on their subphenotypes. It is crucial to acknowledge the population-specific nature of EHR databases and the resulting LCA models. Thus, population-specific subphenotyping models should be developed, instead of one single model applying to all populations. Future studies should also be conducted to evaluate if personalised treatment based on LCA-derived subphenotypes is clinically useful in reducing the risk of CVE in patients with hip fracture.

## Methods
The protocol of this study was approved by the Institutional Review Board in Hong Kong for the use of CDARS database (Reference

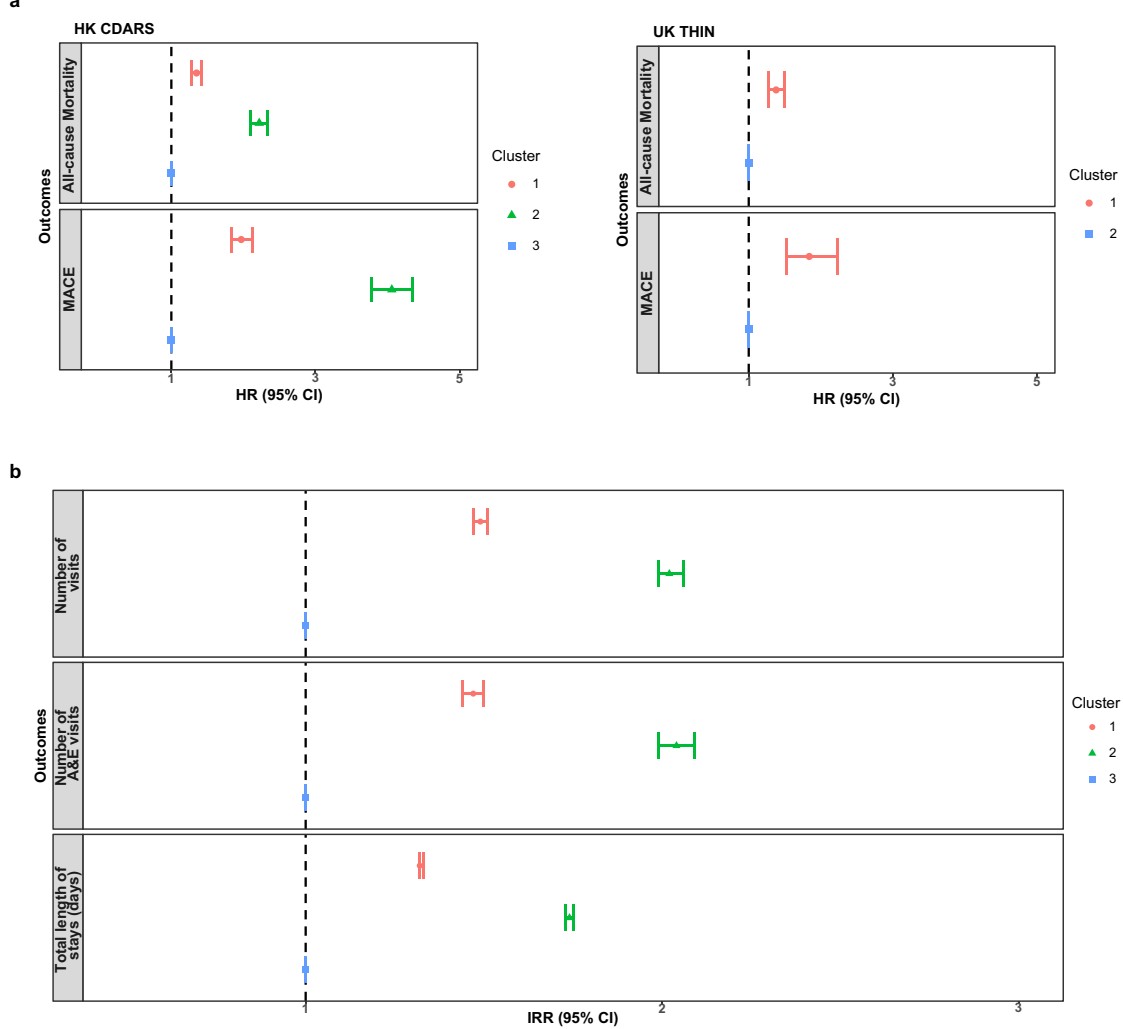

**Fig. 1 | The association between hip fracture subphenotypes and outcomes of interest. a** All-cause mortality and MACE. **b** Hospital outcomes (HK CDARS only). Data are presented as adjusted HR and associated 95% CI derived from the Cox proportional regression for All-cause mortality, and competing risk regression for MACE. For the hospital outcomes, data are presented as adjusted IRR and associated 95% CI derived from Poisson regression. The HR and IRR are indicated by the central symbol, and the 95% CI are indicated by the error bar. The dashed line represents a HR/IRR of 1. Cluster 3 in HK CDARS and Cluster 2 in UK THIN are the reference groups. $n = 78,417$ independent patients in HK CDARS ($n = 16,762$ in Cluster 1; $n = 9860$ in Cluster 2; $n = 51,795$ in Cluster 3). $n = 27,948$ independent patients in UK THIN ($n = 4966$ in Cluster 1; $n = 22,982$ in Cluster 2). MACE major adverse cardiovascular events, HR hazard ratios, CI confidence interval, A&E accident and emergency, IRR incidence rate ratios.

Number: UW22-076), and the Scientific Review Committee (SRC) in the UK for the use of THIN database (Reference Number: 23SRC001). IQVIA Medical Research Data (IMRD) incorporates data from THIN, a Cegedim Database. Reference made to THIN is intended to be descriptive of the data asset licensed by IQVIA. The research was conducted in compliance with all ethical regulations. Informed consents were exempted as all patients were non-identifiable in this study.

## Overview

First, we used the LCA to subphenotype the hip fracture cohorts from Hong Kong and the UK, derived from electronic health records (EHRs). Second, survival analysis was conducted to investigate the associations between the identified hip fracture subphenotypes and MACE, all-cause mortality and health service utilisation outcomes. Finally, the temporal association between hip fracture and MACE was explored using two approaches: the between-individual comparison approach of competing risk regression and the within-individual comparison approach of SCCS.

## Data sources

Two independent hip fracture patient cohorts were constructed using EHRs from the CDARS database in Hong Kong and THIN database in the UK.

CDARS is used for audit and research purposes in the public healthcare system managed by the Hospital Authority in Hong Kong. It captures clinical information of patients including demographics, diagnoses, prescriptions, procedures, laboratory tests and mortality from 43 public hospitals and 122 outpatient clinics in Hong Kong. CDARS covers >80% of hospital admissions in HK, and around 98% of hip fracture cases in HK were admitted to hospitals under HA[31]. THIN is a primary care EHR database covering about 6% of the population with more than 700 general practitioner practices in the UK, and has been shown to be representative of the general UK population in terms of demographics, chronic condition prevalence and mortality rates[32].

The International Classification of Diseases (ICD) code system is used by healthcare professionals to record diagnosis data in HK CDARS, while the Read code system is used by general practitioners to record diagnoses in the UK THIN. The data quality of both databases

**Table 5 | Results of the SCCS analysis for the temporal association of hip fracture subphenotypes with the risk of MACE in HK CDARS**

| Hip fracture exposure window, d | Cluster 1 | | Cluster 2 | | Cluster 3 | | Whole | |
|---|---|---|---|---|---|---|---|---|
| | No. of events | IRR[a] (95% CI) | No. of events | IRR[a] (95% CI) | No. of events | IRR[a] (95% CI) | No. of events | IRR[a] (95% CI) |
| | 2005 | | 1695 | | 2666 | | 6366 | |
| 1–60 d | 146 | 1.84 (1.54, 2.20) | 114 | 1.99 (1.62, 2.45) | 308 | 2.05 (1.81, 2.33) | 568 | 1.93 (1.76, 2.11) |
| 61–120 d | 70 | 0.9 (0.70, 1.15) | 86 | 1.53 (1.22, 1.93) | 211 | 1.39 (1.20, 1.62) | 367 | 1.25 (1.12, 1.39) |
| 121–180 d | 62 | 0.81 (0.63, 1.05) | 63 | 1.15 (0.89, 1.50) | 161 | 1.01 (0.86, 1.20) | 286 | 0.97 (0.86, 1.10) |
| Baseline period[b] | 1727 | 1 (Reference) | 1432 | 1 (Reference) | 1986 | 1 (Reference) | 5145 | 1 (Reference) |

*MACE* major adverse cardiovascular events, *d* day, *IRR* incidence rate ratios, *CI* confidence interval.
[a]IRR, incidence rate ratio adjusted for age by quintiles.
[b]Baseline period indicates 366 days before hip fracture plus 181–732 days after hip fracture.

has been validated for the purpose of epidemiological research[33,34]. Both databases have been used to conduct population-wide observational studies associated with hip fracture in Hong Kong[3,6,35–37] and the UK[38–40].

## Study design
In the CDARS, patients aged ≥65 years admitted to hospital with newly diagnosed hip fractures between January 1, 2005 and December 31, 2020 were identified using the ICD-9 codes of 820.XX. Only the patients who survived until hospital discharge were included and the index date was defined as the hospital discharge date.

In THIN, Read codes were used to identify patients aged ≥65 years with newly diagnosed hip fractures between January 1, 2005 and December 31, 2018. Unlike CDARS which is a hospital EHR database, THIN is implemented in a primary care setting, and therefore the hip fracture event date recorded by the general practitioners was used as index date instead of hospital discharge dates.

## Clustering variables and variables harmonisation
Twenty-two variables were chosen as candidate variables for clustering (Supplementary Table 12). The list of variables includes the demographic variables of sex and age, and a set of 20 individual baseline diagnosis variables associated with hip fracture identified in previous studies[3,6]. These diagnosis variables were identified from the previous literature with consideration including biological plausibility, and included disease classes, e.g., cardiovascular diseases, respiratory-related diseases, and endocrine and metabolic disorders. The baseline status of each diagnosis variable was defined as the presence of a diagnosis record in the 5-year period prior to the index date. The sex variable was determined based on the information health professionals recorded in HK CDARS and UK THIN. As highly correlated clustering variables could lead to bias in the clustering results, the correlations among cluster variables were computed by Spearman's rank correlation coefficients and a correlation coefficient of >0.5 represents a pair of highly correlated variables.

To harmonise the identification of variables in the CDARS and THIN databases, we used the HDR UK Phenotype Library[41], which offers a standardised approach for mapping diagnosis codes across different coding systems in EHR systems (e.g., ICD-9 and Read codes for this study). This open-source resource provides the diagnosis codes for a comprehensive list of phenotypes. Furthermore, the UK Biobank's resource 592 "Clinical coding classification systems and maps" was utilised to translate the ICD-9 codes used in the HK CDARS cohort to Read codes in the THIN cohort to define the baseline diagnosis variables[42].

## Outcome ascertainment
All the outcomes of interest in this study reflected the prognosis of hip fracture patients. The primary outcomes were all-cause mortality and MACE, which were defined by stroke, MI, or hospitalisation due to heart failure (HF) through an A&E visit in HK CDARS[43] (Supplementary Table 13). In UK THIN, MACE was defined by stroke or MI. HF hospitalisation was unavailable in the primary care database THIN. All the outcomes were defined as the incidence within 180 days after the index date, since we previously observed that the temporal association was mainly confined to the first 180 days after hip fracture[6]. The secondary outcomes were on health service utilisation, including the number of hospital visits, the number of A&E visits, and the total length of hospital stays (in days) within the 180-day period after the index date. Similarly, the secondary outcomes were only available in the hospital-based HK CDARS cohort, but not in the primary care-based UK THIN cohort.

## Latent class analysis
LCA was applied to identify hip fracture subphenotypes, which represent patient subgroups sharing similar characteristics. In LCA, class membership probabilities were computed for each patient, allowing for the assignment of each patient to the cluster with the highest membership probability. One notable advantage of LCA is its nature as a model-based clustering method, enabling the calculation of model fit statistics to determine the best model with the optimal number of clusters[9]. The availability of the statistical assessment on clustering solutions enhances the robustness of LCA compared to the conventional clustering algorithms such as k-means clustering, and was shown to be the optimal clustering algorithm for health data among various commonly used clustering algorithms[12].

Following the design of conventional machine learning studies[44], the CDARS cohort was randomly split into the training (70%) and internal test (30%) sets, and the entire THIN dataset served as an independent external validation cohort (Supplementary Fig. 1). The number of clusters was the crucial parameter for the optimal clustering solution. Bayesian information criterion (BIC), average silhouette width (ASW), and integrated complete likelihood (ICL) were used for model comparison. A clustering model with a lower BIC, higher ASW or higher ICL is preferred as these are indications of a better model fit[45]. Using the HK CDARS training set, iterations of LCA were run from 1 to 10 clusters, with BIC, ASW and ICL computed at each iteration. The BIC, ASW and ICL values of the ten models were then compared to determine the model with the best performance and the corresponding optimal number of clusters. This process was independently repeated on both the HK CDARS internal test set and the UK THIN external dataset to assess the reproducibility and generalisability of the LCA solution.

Descriptive statistics were computed to report and compare the baseline characteristics of each LCA-generated cluster. Continuous variables were summarised by mean and standard deviation (SD), and categorical variables were summarised by frequency and percentage (%). Standardised mean differences (SMD) were computed to evaluate the differences between the clusters in terms of the clustering

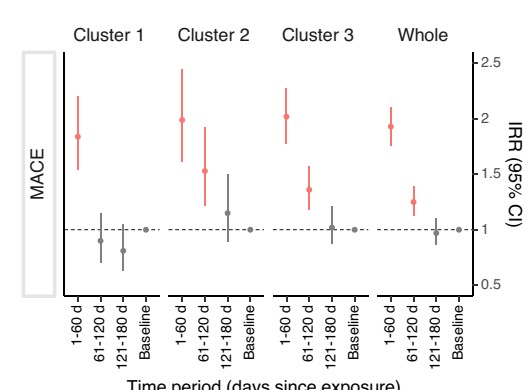
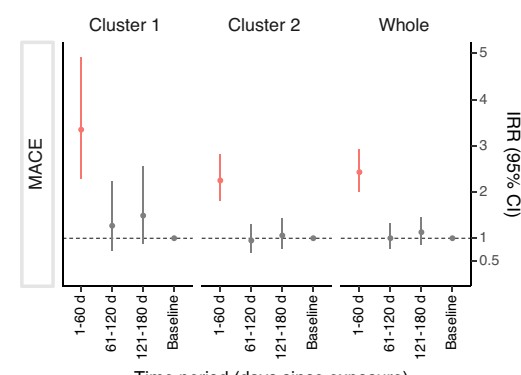

**Fig. 2 | Temporal association of hip fracture subphenotypes with the risk of MACE in the within-individual analysis using SCCS. a** HK CDARS; **b** UK THIN. Data are presented as estimated IRR and associated 95% CI. The IRR are indicated by the central symbol, and the 95% CI are indicated by the error bar. The dashed line represents an IRR of 1. MACE major adverse cardiovascular events, d day, IRR incidence rate ratios, CI confidence interval. IRR, incidence rate ratio adjusted for age by quintiles. Baseline period indicates 366 days before hip fracture plus 181 to 732 days after hip fracture.

**Table 6 | Results of the SCCS analysis for the temporal association of hip fracture subphenotypes with the risk of MACE in UK THIN**

| Hip fracture exposure window, d | Cluster 1 | | Cluster 2 | | Whole | |
|---|---|---|---|---|---|---|
| | No. of events | IRRª (95% CI) | No. of events | IRRª (95% CI) | No. of events | IRRª (95% CI) |
| | 382 | | 844 | | 1226 | |
| 1–60 d | 38 | 3.35 (2.28, 4.92) | 99 | 2.25 (1.81, 2.81) | 137 | 2.43 (2.01, 2.94) |
| 61–120 d | 14 | 1.27 (0.72, 2.23) | 42 | 0.95 (0.69, 1.31) | 56 | 1.00 (0.76, 1.32) |
| 121–180 d | 16 | 1.49 (0.88, 2.55) | 47 | 1.06 (0.78, 1.43) | 63 | 1.13 (0.87, 1.46) |
| Baseline periodᵇ | 314 | 1 (Reference) | 656 | 1 (Reference) | 970 | 1 (Reference) |

*MACE* major adverse cardiovascular events, *d* day, *IRR* incidence rate ratios, *CI* confidence interval.
ªIRR, incidence rate ratio adjusted for age by quintiles.
ᵇBaseline period indicates 366 days before hip fracture plus 181–732 days after hip fracture.

variables. SMD values of 0.2–0.5, >0.5–0.8, and >0.8 were considered to have small, medium, and large differences between groups, respectively[46]. To further assess cluster similarity between training and test sets, bivariate Pearson's correlation coefficients were computed based on the conditional probabilities for all the clustering variables resulting from the LCA solution. A correlation coefficient closer to one represents high association between two clusters.

### Survival analysis

After validating the robustness of the LCA solutions on both HK CDARS training and test sets, the two sets were merged to increase power for all the subsequent prognosis analyses. Cox proportional hazard model was used to investigate the associations between the identified hip fracture subphenotypes and all-cause mortality. Given that the 180-day mortality following hip fracture is high, the competing risk regression model was used to evaluate the association between hip fracture subphenotypes and MACE, with death being the competing event. The secondary outcomes of the number of hospital visits, number of A&E visits and the length of hospitalisation were investigated using Poisson regression. Age and sex were adjusted in all the models. Hazard ratios were computed from the Cox and competing risk regression models, while incidence rate ratios were computed from the Poisson regression models. For all the models, 95% CIs were computed, and statistical significance was defined as a 2-sided *p*-value < 0.05.

Sensitivity analyses were conducted: (1) excluding patients with MACE within 30 days after the index date since such event could potentially represent a post-operative cardiac complication or a delayed coding of a cardiac event just before a hip fracture; (2) refining the MACE outcome to MACE hospitalisations admitted through an A&E visit due to any of the events stroke, MI or HF, which further addressed the potential issue of a pre-existing CVE being coded as a post-fracture diagnosis record. MACE hospitalisation served as a stringent definition of the cardiac outcome, as only a new hospitalisation event due to MACE was regarded as a post-fracture MACE. Another sensitivity analysis was also performed using a more conventional definition of MACE in HK CDARS, which included only acute myocardial infarction and stroke[47]. Furthermore, stratified analyses were conducted to evaluate the associations between the hip fracture subphenotypes and the outcomes of MACE and mortality, across sex, age, and the types of surgical treatment received (internal fixation and partial hip replacement).

### Temporal analysis

The temporal association between the hip fracture subphenotypes and MACE was explored using competing risk regression and SCCS. Since our previous study demonstrated an immediate risk of MACE in hip fracture patients when compared to a healthy control group[6], the temporal association was investigated by comparing the post-fracture MACE risk across clusters, following the competing risk regression-based design with hazard ratios computed at 90, 180, 270 and 366 days after index date.

In the SCCS analysis, only patients with both exposure (hip fracture) and outcome (MACE) and with two years of follow-up data available after the index date, were included in the analysis. Patients with the outcome of interest were followed from 366 days before the hip fracture to 732 days after it (Supplementary Fig. 7). The risk periods were 1–60, 61–120 and 121–180 days. On the individual level,

all the fixed confounders were controlled implicitly; the risk of the event during the risk periods was compared to that of the control periods, where the IRR was constant and equalled 1.0. The null hypothesis, where IRR = 1.0, implied that the event rate remained constant during the entire observation period and was not affected by having a hip fracture. Since we cannot determine whether the event that happened on day 0 was after or before the exposure based on the diagnosis codes and date, we excluded those with events on day 0. All the analyses were also performed in each of the hip fracture subphenotypes.

Only the first event of MACE was included in the SCCS analysis. The SCCS design assumes an event does not influence the subsequent period of observation or subsequent exposures. However, it is known that cardiac events are associated with an increased risk of hip fractures[20,48]. To address this issue, we adopted the modified SCCS model with event-dependent exposure[49]. Similarly, SCCS assumes the occurrence of the outcome event does not shorten the observation period[50]. We therefore excluded those censored before the end of the observational period. The IRRs were adjusted for age at the exposure by quintile age groups. The robustness of the SCCS analysis was evaluated by (1) using shorter exposure intervals (i.e., 1–30, 31–60, 61–90, 91–120, 121–150 and 151–180 days after the exposure), given that MACE occurred in the period 1–30 days after the exposure could be attributed to post-operative cardiac events[24] and (2) considering only the post-hip fracture period as the baseline period.

All the statistical analyses were conducted in R[51]. The R package poLCA[52] (version 1.6.0.1) and the R script published by Lezhnina and Kismihok[45] (2022) were used to run the LCA, the package "cmprsk" (version 2.2-11) was used to run the competing risk regression[53], and the package "SCCS" (version 1.6) was used for the SCCS analyses[50].

### Reporting summary
Further information on research design is available in the Nature Portfolio Reporting Summary linked to this article.

## Data availability
The data used in this study cannot be shared with the public due to third-party use restrictions and patient confidentiality concerns. The HK CDARS EHR database is directly under the control of the Hong Kong Hospital Authority. Local academic institutions, government departments, or non-governmental organisations can apply for access to CDARS data through Hong Kong Hospital Authority Data Sharing Portal (https://www3.ha.org.hk/data). The detailed application procedure can be found at https://www3.ha.org.hk/data/Provision/ApplicationProcedure. The UK THIN, a Cegedim EHR Database, is licensed by IQVIA. It is available for researchers from academic, public health, research establishment, charitable, commercial and regulatory bodies through purchase. Information on IQVIA Medical Research Data (IMRD) which incorporated the UK THIN data, can be found at https://www.iqvia.com/locations/united-kingdom/information-for-members-of-the-public/medical-research-data. Applications to access the UK THIN data can be made via https://www.the-health-improvement-network.com/.

## Code availability
This study did not generate any new algorithm/model. Statistical analyses were performed using R software (version 4.3.0; R Foundation for Statistical Computing, Vienna, Austria) through R packages provided in the 'Methods'. The codes used in the analyses are available from the corresponding author.

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

## Acknowledgements

This work was supported by AIR@InnoHK administered by Innovation and Technology Commission. This study received funding support from the Health and Medical Research Fund, Food and Health Bureau, The Government of the Hong Kong Special Administrative Region (Reference 18192451) and Seed Fund for PI Research, University of Hong Kong (Reference 202111159128).

## Author contributions

W.W.Q.H: Conceptualization; data curation; formal analysis; investigation; visualization; writing—original draft; writing—review and editing; methodology; software. X.Z: Conceptualization; formal analysis; investigation; visualization; writing—original draft; writing—review and editing; methodology; software. C.W.S: Conceptualization; writing—review and editing. K.C.B.T: Resources; project administration. I.C.K.W: Resources; project administration; supervision. W.C.Y.L: Resources; methodology. C.L.C: Conceptualization; resources; project administration; supervision; writing—review and editing; methodology; funding acquisition.

## Competing interests

The authors declare no competing interests.
