## [Peer Review File · Nature Communications]

REVIEWER COMMENTS

Reviewer #1 (Remarks to the Author):

Paper Review:

In this study, the authors identify sub-phenotypes among hip fracture patients in two large data sets, HK CDARS and UK THIN, based on categorical factors potentially associated with or measuring some aspect of cardiovascular health. Given that cardiovascular diseases are the primary cause of death in the adult population, this goal is useful for aiding in more directed care of hip fracture patients as the authors note in their discussion. To identify the clusters in the data sets based on these factors, the authors employ polytomous variable latent class analysis (LCA), which is a well-supported method for unsupervised clustering problems. The number of clusters is obtained via tuning; there are three in HK CDARS and two in UK THIN. The authors report the characteristics of each cluster and evaluate health outcomes within them using survival analysis (discussed in much greater detail below), demonstrating that they have picked out distinct sub-phenotypes relative to cardiovascular health and prognoses.

All in all, I think that the article is well-written and includes rigorous and reproducible statistics from viable data sets with a noteworthy goal of identifying sub-phenotypes of hip fracture patients susceptible to certain negative health events. The methods with sensitivity analyses, results, and conclusions are solidly presented. However, the authors need to further justify their motivation for conducting each of the survival analyses for MACE after clustering into sub-phenotypes as described in the following comments.

Major Comments:

1. For the survival analysis, the primary outcomes are all-cause mortality and MACE, and the secondary outcomes are ones related to hospitalization. The authors first consider the risk of these events in the less healthy cluster(s) as compared to the healthiest cluster that were identified using LCA, which they refer to as a between-individual framework. Most compellingly, they estimate that the hazard ratios for all-cause mortality are larger than one (significantly so) in the less healthy cluster(s) relative to the healthiest cluster using Cox models adjusted for age and sex. This observation on its own supports their thesis that there are sub-phenotypes within hip-fracture patients that are differentially vulnerable. Their emphasis in the temporal stage of the between-individual survival analysis is on the event MACE. They use a Fine and Gray competing risk regression model adjusted for age and sex with time-dependent effects to compare clusters where the

competing event is death. This is a valid statistical method and is clearly described in the paper, but the analysis feels self-evident. The greatest imbalances between the less healthy cluster(s) as compared to the healthiest cluster are due to factors that are directly related to cardiovascular health e.g., coronary heart disease or congestive heart failure. We should of course expect a comparatively higher risk of MACE within the less healthy clusters. In particular, the UK THIN data set is only separated into two clusters - one of which is overwhelmingly less healthy than the other vis-à-vis those strong cardiovascular health indicators. There appears to be a benefit in identifying these latent classes relative to mortality, but in the case of MACE, are we just saying that hip fracture patients with worse cardiovascular problems are at higher risk of adverse cardiovascular events? As written in the discussion, “such findings could be driven by the higher prevalence of prevalent cardiac events in these clusters and other potential unmeasured confounders.” I think that further clarification as to why the authors decide to compare (and emphasize) MACE risk across these clusters would be helpful to include, where they might report as in Seymour et al. (2019, Citation 11) “several analyses...conducted to ensure the phenotypes were not simply recapitulations of more traditional clinical groups”. Even as it pertains to mortality, the authors may want to point out that the less healthy sub-phenotypes are not ones that would readily be classified as higher risk.

2. Their second approach to analyzing MACE temporally is the self-controlled case series (SCCS) method, which they call a within-individual approach. Using this method, only hip fracture patients who experience the event are included, where there are various risk periods following exposure (hip fracture) including times when the patient is not at risk as reference. In this way, patients are used as their own control accounting for time-fixed confounders. Once again, their analysis is methodologically valid (addressing notably event-dependent exposure) as well as conveyed with clarity. That said, it does target something different than the previous temporal analysis, which really views the cluster membership as the exposure as opposed to the hip fracture itself. Therefore, it may be helpful to clarify this distinction and to explain the purpose of performing the SCCS analysis after clustering. Indeed, the reported results show that the cluster-level IRRs have relatively comparable temporal trends and are highest with similar values 1-60 days after a hip fracture. These cluster-specific patterns match the overall pattern in IRR as well. In the results section, the authors emphasize that “significantly higher age-adjusted IRRs of MACE at 1-60 days were observed after hip fracture”, which makes it seem that their main point here is not the clustering into sub-phenotypes but the temporal effect of the exposure on its own.

Further Considerations:

1. How is a single Pearson's correlation coefficient obtained for cluster comparison as in Supplementary Table 2 (comparing training to testing) or in lines 120-123 (comparing clusters across data sets)? I think this is explained with respect to estimated conditional cluster probabilities as per LCA, but could the authors elaborate on this statistic a bit?

2. How are all three model performance measures, BIC, ASW, and ICL, aggregated to select the model with the optimal number of clusters?

3. Is there some measure of cluster membership uncertainty?

4. How are missing data handled?

Reviewer #2 (Remarks to the Author):

- In this manuscript, the authors describe a machine learning protocol identifying clusters of comorbidities among individuals with hip fracture hospitalizations. The objective of the manuscript is to examine the association of so identified patient groups with the risk of adverse short-term cardiovascular events. Not surprisingly, the authors observed a positive association of pre-existing cardiovascular morbidities, such as heart failure, with major adverse cardiovascular event risk. The novelty of this finding depends very much on clinical practice. It would seem logical that the management of hip fracture patients would include, in addition to the fracture itself, also attendant comorbidities. Such information is readily available in clinical charts. The premise of this manuscript is therefore not very clear.
- The authors do not include information on the severity of the hip fractures or anything related to hip replacement. Further, analyses are not stratified by age categories or sex, which would provide additional information pertaining to likelihood of cardiovascular outcomes.
- The manuscript includes language that is suggestive of a causality between hip fractures and cardiovascular outcomes. This is likely not intentional, however, whereas it is appropriate to discuss the association of comorbidity clusters with cardiovascular outcomes among hip fracture patients, it is not appropriate to make a statement about the association of hip fractures with cardiovascular outcomes (e.g. page 8, line 169), as the causal mechanism is not there. This is further incorrect, given that the study population included exclusively patients with hip fractures.
- I had a difficult time understanding the findings presented in Figure 3. It would appear that each of the identified clusters was associated with MACE at 60 days post-admission. The maintenance of this risk within an extended 180-day window does not imply a dose relationship, as the figure suggests. Partitioning the risk into time periods, as presented in Figure 3, necessitates a re-adjustment of the denominators, which does not appear to have been done. The relevance of the identified clusters in terms of MACE risk is not clear from this figure.

Reviewer #3 (Remarks to the Author):

The manuscript by Hsu and colleagues entitled "Unveiling Unique Clinical Phenotypes of Hip Fracture Patients and the Temporal Association with Cardiovascular Events in Hong Kong and the United Kingdom: A Retrospective Study" describes findings from two prospective cohort studies of people ≥ 65 years with hip fractures, the HK CDARS hip fracture cohort (n=78,417) and the UK THIN cohort (n=27,968) investigating the use of latent class analysis to identify clusters or sub phenotypes of hip fracture patients based and investigate their association with incident cardiovascular events. The authors found that there were three sub phenotypes (clusters) in the HK CDARS and two sub phenotypes in the UK THIN cohorts. The sub phenotypes with atherosclerotic disease (cluster 1) and that with heart failure (cluster 2) were at 2-4 times higher relative hazard for cardiovascular events. With this risk being most evident in 1-60 days post hip fracture. This is noteworthy as post-hip fracture risk of CVD events is not well described in literature or recognized more widely as a CVD risk factor. The manuscript is very well written, is original and provides enough details to be reproduced.

Strengths are that the study has utilised two large hip fracture cohorts and have carefully designed and undertaken this work. The results and findings are of great interest and potentially clinically meaningful.

Limitations of the study are that the results section is quite difficult to follow, and the discussion is quite cursory in parts i.e., does not discuss how these findings may be operationalised in a clinical setting such as hospitals or fracture liaison services and does not discuss how hip fracture is a non-traditional gender-specific cardiovascular disease risk factor and how the findings from this manuscript may contribute to ongoing efforts to bridge improve CVD outcomes in women.

Discussion on how the hip-fracture 60 day and 12-month MACE events rates compare to the post-myocardial infarction 60 day and 12-month MACE and mortality events rates would be informative for the readers to allow comparison for the observed gradient of risk between these clusters.

The cohorts' clinical characteristics and particularly heart failure, hypertensive disease, diabetes, osteoporosis and renal diseases are very different between the cohorts, and the clustering solutions were similarly different. This makes interpretation and operationalisation of the findings difficult. Greater discussion of this issue and the next steps needed are warranted.

The composite components for the MACE outcome are not commonly used (Bosco, Elliott, et al. "Major adverse cardiovascular event definitions used in observational analysis of administrative databases: a systematic review." *BMC Medical Research Methodology* 21.1 (2021): 1-18). In the HK CDARS the increase in the Heart failure events did appear substantially greater than for the other components of MACE. This outcome should be justified by a reference or previous work. It would also be good to compare to a more commonly used definitions such as AMI, stroke and all-cause mortality.

Minor

Figure 3 - appears redundant give Table 1.

Abstract: Some quantification of the “higher risks of major adverse cardiovascular events” should be included in the abstract.

Abstract: The study abstract aims sentence “This study aimed to identify subphenotypes of hip fracture patients and investigate their association with incident CVEs in Hong Kong (HK) and the United Kingdom (UK) populations.” fails to mention all-cause mortality and health service utilisation as outcomes, despite reporting the results of these outcomes.

REVIEWER COMMENTS

Reviewer #1 (Remarks to the Author):

Paper Review:

In this study, the authors identify sub-phenotypes among hip fracture patients in two large data sets, HK CDARS and UK THIN, based on categorical factors potentially associated with or measuring some aspect of cardiovascular health. Given that cardiovascular diseases are the primary cause of death in the adult population, this goal is useful for aiding in more directed care of hip fracture patients as the authors note in their discussion. To identify the clusters in the data sets based on these factors, the authors employ polytomous variable latent class analysis (LCA), which is a well-supported method for unsupervised clustering problems. The number of clusters is obtained via tuning; there are three in HK CDARS and two in UK THIN. The authors report the characteristics of each cluster and evaluate health outcomes within them using survival analysis (discussed in much greater detail below), demonstrating that they have picked out distinct sub-phenotypes relative to cardiovascular health and prognoses.

All in all, I think that the article is well-written and includes rigorous and reproducible statistics from viable data sets with a noteworthy goal of identifying sub-phenotypes of hip fracture patients susceptible to certain negative health events. The methods with sensitivity analyses, results, and conclusions are solidly presented. However, the authors need to further justify their motivation for conducting each of the survival analyses for MACE after clustering into sub-phenotypes as described in the following comments.

Major Comments:

1. For the survival analysis, the primary outcomes are all-cause mortality and MACE, and the secondary outcomes are ones related to hospitalization. The authors first consider the risk of these events in the less healthy cluster(s) as compared to the healthiest cluster that were identified using LCA, which they refer to as a between-individual framework. Most compellingly, they estimate that the hazard ratios for all-cause mortality are larger than one (significantly so) in the less healthy cluster(s) relative to the healthiest cluster using Cox models adjusted for age and sex. This observation on its own supports their thesis that there are sub-phenotypes within hip-fracture patients that are differentially vulnerable. Their emphasis in the temporal stage of the between-individual survival analysis is on the event MACE. They use a Fine and Gray competing risk regression model adjusted for age and sex with time-dependent effects to compare clusters where the competing event is death. This is a valid statistical method and is clearly described in the paper, but the analysis feels self-evident. The greatest imbalances between the less healthy cluster(s) as compared to the healthiest cluster are due to factors that are directly related to cardiovascular health e.g., coronary heart disease or congestive heart failure. We should of course expect a comparatively higher risk of MACE within the less healthy clusters. In particular, the UK THIN data set is only separated into two clusters - one of which is overwhelmingly less healthy than the other vis-à-vis those strong cardiovascular health

indicators. There appears to be a benefit in identifying these latent classes relative to mortality, but in the case of MACE, are we just saying that hip fracture patients with worse cardiovascular problems are at higher risk of adverse cardiovascular events? As written in the discussion, “such findings could be driven by the higher prevalence of prevalent cardiac events in these clusters and other potential unmeasured confounders.” I think that further clarification as to why the authors decide to compare (and emphasize) MACE risk across these clusters would be helpful to include, where they might report as in Seymour et al. (2019, Citation 11) “several analyses...conducted to ensure the phenotypes were not simply recapitulations of more traditional clinical groups”. Even as it pertains to mortality, the authors may want to point out that the less healthy sub-phenotypes are not ones that would readily be classified as higher risk.

Response: Thank you for your comments and providing the opportunity to elaborate on the motivations and implications of our analyses.

One of our major objectives was to gain a deeper understanding of the post-fracture CVE risk in hip fracture patients. In line with the objective, we set up our study plan before conducting the LCA. We planned to first identify the subphenotypes of hip fractures, and then conduct a survival analysis following LCA to quantify the post-hip fracture CVE risk over time. The objective was to determine if there are differences in prognosis outcomes across subphenotypes.

In a previous LCA analysis of diabetic patients (Seng JJB, et al. Differential Health Care Use, Diabetes-Related Complications, and Mortality Among Five Unique Classes of Patients With Type 2 Diabetes in Singapore: A Latent Class Analysis of 71,125 Patients. Diabetes Care. 2020 May;43(5):1048-1056.), five subphenotypes were identified, and only one of them (class 5) showed significant enrichment in CVEs. The prevalence of CVEs in the other classes were close to or below the mean percentage. In contrast, our analysis of HK CDARS identified two subphenotypes rather than one single group showing significant enrichment in CVEs. Cerebrovascular diseases and arterial disease (vascular diseases) were enriched in Cluster 1, while coronary heart disease, congestive heart failure, arrhythmia and conductive disorders (cardiac diseases) were enriched in Cluster 2. These findings suggest that, unlike diabetes, which showed a strong association with CVEs collectively, hip fracture exhibited a differential association with multiple cardiac and vascular diseases.

In addition, LCA was conducted without presumptions regarding which of the clustering variables would be particularly important in the subphenotyping process. Nevertheless, it revealed the importance of baseline CVEs.

Last but not least, CVE risk management in hip fracture care receives little emphasis in clinical practice, despite an association between baseline CVE and increased post-fracture CVE risk. Our results highlighted the critical need to recognise the heterogeneity among hip fracture patients and the elevated CVE risk, particularly the temporal risk, after hip fracture.

We provide the clarification in the introduction and discussion:

[Introduction, page 4: last sentence; page 5: paragraph 1]

“...Therefore, in this study, we first utilised the unsupervised LCA to subgroup hip fracture patients who share common clinical characteristics in two extensive hip fracture cohorts: one from Hong Kong (HK; N=78,417) and the other from the United Kingdom (UK; N=27,948). Subsequently, to uncover variations in long-term outcomes for hip fracture patients based on their subphenotypic classification, conventional survival analyses (between-individual comparison) and self-controlled case series (SCCS; within-individual comparison) were conducted. The analyses aimed to quantify the risk of CVE-related outcomes, all-cause mortality, and health service utilisation following hip fracture and to evaluate potential differences across the identified subphenotypes. Unlike the conventional survival analyses comparing the prognosis risk between individuals, SCCS analyses focused only on hip fracture patients who experienced the prognostic events with each patient serving as their own control, which effectively controlled for time-fixed confounders and between-individual differences¹³. Consequently, SCCS analyses enabled the detection of temporal patterns within each LCA-derived cluster and facilitated the identification of potential differences in temporal patterns across the clusters, which cannot be captured by conventional survival analyses alone. Overall, the conventional survival analysis and SCCS are complementary approaches that provide a comprehensive understanding of the prognosis risk and temporal associations in the identified subphenotypes.”

[Discussion, page 17-18]

“This study has important implications. Notably, the LCA results revealed subphenotypes with pronounced baseline CVE profiles across two independent population-based cohorts, without any preconceived assumption about CVE history being particularly important among the clustering variables. Although current clinical guidelines highlight the importance of perioperative evaluation and optimisation for patients with a high cardiac risk^{26 27}, there is a noticeable lack of emphasis on CVE risk management in the specific context of hip fracture care. For example, current heart failure management guidelines rarely cover the management of osteoporosis and hip fracture risk in heart failure patients^{28 29}. Our findings supported by the previous studies emphasise the importance of taking measures to mitigate the risk of hip fracture among heart failure patients, given the potential poor prognosis after hip fracture. Notably, our temporal analysis showed an immediate increased risk of MACE after hip fracture, with the incidence rates of Cluster 2 in HK CDARS and Cluster 1 in UK THIN being particularly high. These findings underscore the importance of prompt CVE risk management especially in these high-risk patient groups.

The elevated CVE risk after hip fracture is inadequately documented in the literature, along with the association between pre-existing CVE and the subsequent risk of CVEs in hip fracture patients, especially considering multimorbidity is common in the older hip fracture patient population.³⁰ Our study addressed this gap by quantifying the associated risks, and the elevated risk of MACE within 60 days post-hip fracture across all subphenotypes highlighted again the

critical need for integrating CVE management into the broader hip fracture care strategy within the first year after hip fracture. Using the relatively healthy cluster as the reference group, this comparative framework allowed us to evaluate the relative prognosis of each unique comorbidity cluster, thereby laying the groundwork for more personalised management strategies and prioritisation of healthcare resources.

Hip fractures are often managed as a homogenous condition, yet the current study explored the heterogeneity among hip fracture patients and reported variations in mortality, CVE risk and hospital utilisation outcomes across subphenotypes. This variation emphasises the need to advance from conventional stratifications, such as those relying on a single baseline condition or demographic factor, to a more holistic, multimorbidity-focused approach in classifying patients, as facilitated by LCA. ...”

In addition, we conducted an additional comparative analysis to clarify that the hip fracture subphenotypes identified were not simple recapitulations of traditional clinical groups. This analysis (suggested by another reviewer) involved comparing 60-day and 12-month MACE and mortality rates between the hip fracture subphenotypes and a reference MI patient cohort from CDARS. This allowed a comparison of the event rate gradients (from 60-day to 12-months) between the subphenotypes identified by LCA and a reference group of traditional CVD patients. Below is the new section added to address the observed differences:

[Results, page 10: paragraph 2]

“Furthermore, to verify the hip fracture subphenotypes were not direct replications of conventional cardiovascular disease (CVD) groups, a comparative analysis was conducted using a reference CDARS cohort composed of patients with myocardial infarction (MI). In this analysis, the 60-day and 12-month event rates for MACE and all-cause mortality for the subphenotypes were compared to that of the reference MI cohort (Supplementary Table 11). The MI cohort revealed expectedly high event rates of MACE and all-cause mortality. Notably, the MI cohort showed a moderate increase (28.48%) in the event rates of MACE (18.89% to 24.27%) and a substantial rise (137.8%) in mortality event rates (11.77% to 27.99%) from the 60-day to the 1-year period. Conversely, the hip fracture cohort, with 3.18% of patients with baseline MI, demonstrated a sharper increase in both outcomes over the same period, with the MACE event rates more than doubling (164.2% increase, from 2.29% to 6.05%) and all-cause mortality rates tripling (255.2% increase, from 4.67% to 16.59%). This trend was consistent across all three clusters within the hip fracture cohort. In particular, Cluster 2 showed the highest absolute event rates among the hip fracture clusters, with the 1-year MACE rate being 10.72%, almost half of that of the MI cohort, even though only 15.04% of patients in Cluster 2 had baseline MI. In addition, the 1-year mortality rate in Cluster 2 surpassed that of the MI cohort (31.35% vs 27.99%).”

Supplementary Table 11. Comparison of the event rates of the hip fracture subphenotypes with the MI reference cohort in HK CDARS.

	Cluster 1	Cluster 2	Cluster 3	Total	MI Reference Cohort
N	16,762	9,860	51,795	78,417	36,933
Baseline MI	3.42%	15.04%	0.85%	3.18%	100%
MACE*					
60-days	3.48%	4.11%	1.56%	2.29%	18.89%
1-year	8.33%	10.72%	4.42%	6.05%	24.27%
All-cause mortality					
60-days	4.53%	9.59%	3.78%	4.67%	11.77%
1-year	17.25%	31.35%	13.57%	16.59%	27.99%

Notes: MACE = Major Adverse Cardiovascular; MI = Myocardial infarction.

*MACE defined by MI and stroke (as heart failure hospitalisation data were not available in the MI Reference Cohort).

[Discussion, page 13, paragraph 2]

“...Such findings could be driven by the higher prevalence of prevalent cardiac events in these clusters and other potential unmeasured confounders. However, the differences in the event rates of MACE and all-cause mortality between the hip fracture subphenotypes and the reference group of MI patients partially supported that the subphenotypes identified by LCA were not simply reiterations of traditional CVD patient groups. ...”

2. Their second approach to analyzing MACE temporally is the self-controlled case series (SCCS) method, which they call a within-individual approach. Using this method, only hip fracture patients who experience the event are included, where there are various risk periods following exposure (hip fracture) including times when the patient is not at risk as reference. In this way, patients are used as their own control accounting for time-fixed confounders. Once again, their analysis is methodologically valid (addressing notably event-dependent exposure) as well as conveyed with clarity. That said, it does target something different than the previous temporal analysis, which really views the cluster membership as the exposure as opposed to the hip fracture itself. Therefore, it may be helpful to clarify this distinction and to explain the purpose of performing the SCCS analysis after clustering. Indeed, the reported results show that the cluster-level IRRs have relatively comparable temporal trends and are highest with similar values 1-60 days after a hip fracture. These cluster-specific patterns match the overall pattern in IRR as well. In the results section, the authors emphasize that “significantly higher age-adjusted IRRs of MACE at 1-60 days were observed after hip fracture”, which makes it seem that their main point here is not the clustering into sub-phenotypes but the temporal effect of the exposure on its own.

Response: Thank you for the comments. In brief, our objective was to examine the temporal patterns within each LCA-derived cluster and determine if distinct patterns were present. These patterns could potentially help explain the temporal association of CVE risk following hip fracture. This analysis is crucial as it provided insights that cannot be obtained through survival analyses, which focused on comparisons between individuals. In HK CDARS, the hazard ratios in the survival analyses only compared

Cluster 2 and Cluster 1 to Cluster 3 (the reference group), which could not capture the specific temporal patterns within each cluster.

Although SCCS results eventually indicated similar temporal patterns across clusters, it is noteworthy that the risk of post-hip fracture MACE remained elevated even among individuals classified as relatively healthy.

Furthermore, we have now added a plot illustrating the incidence rates of post-hip fracture MACE by clusters over time (refer to Supplementary Figure 6 below). This plot also showed similar temporal patterns of incidence rates across clusters, but notably, Cluster 2 in HK CDARS exhibited the highest incidence rate, followed by Cluster 1, while Cluster 3 had the lowest incidence rate. This aligned with the results of between-individual analyses showing an increased risk of MACE in Cluster 2 and 1 when compared to Cluster 3. Overall, although the temporal patterns across clusters were similar, the risk of MACE in each cluster varied. Specifically, Cluster 2 has the highest risk of post-hip fracture MACE.

We clarified the distinction and the purpose of conducting SCCS after clustering in the text.

[Introduction, page 5, paragraph 1]

"...Unlike the conventional survival analyses comparing the prognosis risk between individuals, SCCS analyses focused only on hip fracture patients who experienced the prognostic events with each patient serving as their own control, which effectively controlled for time-fixed confounders and between-individual differences¹³. Consequently, SCCS analyses enabled the detection of temporal patterns within each LCA-derived cluster and facilitated the identification of potential differences in temporal patterns across the clusters, which cannot be captured by conventional survival analyses alone. Overall, the conventional survival analysis and SCCS are complementary approaches that provide a comprehensive understanding of the prognosis risk and temporal associations in the identified subphenotypes."

We agree that the articulation of the results was not closely aligned with the purpose of the SCCS analysis. We modified the results and discussion to the following:

[Results, page 9, paragraph 3]

"...Similar temporal association patterns were observed in all clusters, including the relatively healthy cluster. Specifically, the age-adjusted incidence rate ratios (IRRs) of MACE were found to be the highest and statistically significant at 1-60 days after hip fracture across the three clusters (Cluster 1: IRR 1.84, 95% CI 1.54 to 2.20; Cluster 2: IRR 1.99, 95% CI 1.62 to 2.45; Cluster 3: IRR 2.05, 95% CI 1.81 to 2.33), and the IRRs decreased during the subsequent post-hip fracture risk periods (Fig. 2). The cluster-specific patterns matched the overall temporal pattern of the entire cohort, with the age-adjusted IRR of MACE at 1-60 days after hip fracture being 1.93 (95% CI, 1.76 to 2.11; Fig. 2). Similar results were observed in the UK THIN database, where significantly higher IRRs of MACE were observed at 1-60 days after hip

fracture in the two clusters (Cluster 1: IRR 3.35, 95% CI 2.28 to 4.92; Cluster 2: IRR 2.25, 95% CI 1.81 to 2.81), matching the overall pattern of the entire cohort (IRR 2.43, 95% CI 2.01 to 2.94; Fig. 2).”

[Results, page 10, paragraph 1]

“...In addition, we plotted the incidence rates of MACE within one year following hip fracture (Supplementary Fig. 6). The plots showed similar temporal patterns across clusters, but there were notable differences in the incidence rates across clusters in both HK CDARS and UK THIN, which aligned with the results from competing risk regression and SCCS.”

[Discussion, page 13, paragraph 2]

“...We further conducted SCCS analyses to address the between-person residual confounding and to investigate the temporal patterns across clusters, which cannot be achieved through conventional survival analyses. While the results revealed similar temporal association patterns across clusters, consistently showing the highest risk of MACE within 1-60 days after the hip fracture, it is important to note that the risk of post-hip fracture MACE remained elevated even among individuals classified as relatively healthy. ...”

Supplementary Fig. 6. Incidence Rates for MACE by Cluster in the Hip Fracture Cohorts.

(a) HK CDARS

Incidence Rates for MACE (per 1000 Person-months)

	90 days	180 days	270 days	1 year
Cluster 1	17.47	12.69	10.78	9.67
Cluster 2	38.16	30.76	26.85	24.61
Cluster 3	8.46	6.53	5.71	5.23

(b) UK THIN

Incidence Rates for MACE (per 1000 Person-months)

	90 days	180 days	270 days	1 year
Cluster 1	8.16	6.39	5.60	4.99
Cluster 2	3.94	3.10	2.90	2.68

Further Considerations:

1. How is a single Pearson's correlation coefficient obtained for cluster comparison as in Supplementary Table 2 (comparing training to testing) or in lines 120-123 (comparing clusters across data sets)? I think this is explained with respect to estimated conditional cluster probabilities as per LCA, but could the authors elaborate on this statistic a bit?

Responses: Thank you for the question and we have now added the following section under the current Supplementary Table 3 to elaborate on the computation of the Pearson's correlation coefficients for cluster comparison:

[Under Supplementary Table 3]

“For each cluster identified, latent class analysis computed a set of conditional probabilities associated with the 22 clustering variables considered in this study. These conditional probabilities, also known as item-response probabilities, were estimated through the maximum likelihood estimation process in LCA. The item-response probabilities are the likelihood of observing a specific response given a latent class. For example, a conditional probability of 0.598 for baseline heart failure in cluster 2 in HK CDARS implies that there is a 59.8% chance that a patient classified to cluster 2 has baseline heart failure.

Each cluster could therefore be characterised by a list of 22 conditional probability values (one for each clustering variable), that could be leveraged for cluster comparison. For example, to

assess the similarity between Cluster 1 in the training set and Cluster 1 in the test set, the Pearson's correlation coefficient was computed using their respective lists of 22 conditional probabilities. This process was repeated for each pair of clusters between the training and test sets.

Higher correlation coefficients towards 1 indicate higher degree of similarity between the two clusters being compared (a similar profile regarding the presence of the clustering variables). The results from Supplementary Table 3 showed consistency of cluster characteristics between the corresponding clusters across the training and test sets.”

2. How are all three model performance measures, BIC, ASW, and ICL, aggregated to select the model with the optimal number of clusters?

Responses: Thank you for the question. The optimal number of clusters was selected by referencing from the three model performance measures, combined with the consideration of clinical practicality. The decision was not solely based on the absolute lowest BIC or the highest values of ASW and ICL, but on the point where the three metrics converged to support a robust and clinically meaningful cluster solution. In terms of clinical practicality, a simpler solution with a smaller number of latent classes was preferred, since constructing a large number of subphenotypes with relatively small class sizes makes the interpretation and operationalisation of the LCA results more difficult. Below are the descriptions of the rationales behind the final decisions across cohorts (now added under the current Supplementary Figure 2):

[Under Supplementary Figure 2]

(a) HK CDARS training set: 3-clusters solution

Rationale: From the BIC plot, a distinct inflection point could be observed at approximately 2 to 3 clusters. Representing a marked improvement in model fit. Although BIC showed a marginal decrease with additional clusters beyond the point of three clusters, the gain in model fit did not outweigh the increased complexity. Furthermore, the absolute value of ASW was the highest at 3 clusters, and ICL was also among the highest levels at 3 clusters. While the ICL achieved its absolute peak at 8 clusters, a solution with fewer clusters is preferable for clinical applicability and ease of interpretation. Therefore, a 3-clusters solution was selected as the optimal solution, striking a balance among BIC, ASW, and ICL, while maintaining clinical interpretability.

(b) HK CDARS test set: 3-clusters solution

Rationale: The trajectories of the three model performance metrics for HK CDARS test set resembled that of the training set. Similarly, for BIC, the marked inflection point was observed at approximately 2 to 3 clusters, with no substantial improvement in model fit beyond this range. The absolute value of ASW was again the highest at 3 clusters, and the ICL at 3 clusters was also among the highest levels. Therefore, 3-clusters was again an appropriate point of

agreement across the three metrics and was selected as the final solution.

(c) UK THIN Cohort: 2 clusters solution

Rationale: The BIC plot for the UK THIN cohort identified a marked inflection zone between 2 to 4 clusters. The absolute ASW value was the highest at 2 clusters, while the ICL value peaked at a single cluster, followed by a steady decline in ICL values with additional clusters. The downward trends in both ASW and ICL beyond two clusters suggested that a smaller number of clusters may better capture the data structure. Integrating all the three metrics led to our conclusion of evaluating a 2-clusters solution for the UK THIN cohort.

3. Is there some measure of cluster membership uncertainty?

Responses: Thank you for the suggestion. To assess the uncertainty of cluster membership, we have now retrieved each patient's highest class membership probability from the LCA results. For each patient, this probability quantifies the confidence in the class assignment, with a higher value indicating greater certainty that a patient aligns with a particular class. We have now analysed the distribution of the highest class membership probabilities within each cluster. This analysis involved computing the median, along with the lower and upper quartiles of these probabilities for all subjects within each cluster (Supplementary Table 2). As shown in the table, the medians of the highest class membership probabilities ranged between 0.82 and 0.98 across the clusters, demonstrating a moderately high to very high certainty in the cluster assignment of subjects. The results suggested that most of the class membership were assigned with a high level of confidence.

The following description has been added under the new Supplementary Table 2 to elaborate our approach:

[Supplementary Table 2]

"Class membership probabilities are computed for each subject through LCA. This probability represents the likelihood that a subject would be assigned to a certain class (computed using the subject's profile defined by the clustering variables). For example, within the context of a three-cluster LCA solution, each patient is associated with three class membership probabilities, corresponding to the likelihood of their belonging to each of the three latent classes. These probabilities quantify the confidence in class assignment, with higher values indicating greater certainty that a patient aligns with a particular class. Patients are assigned into the class for which they hold the highest membership probability."

The following was added to the result section to report on cluster membership uncertainty:

[Results, page 6, paragraph 2]

“...To assess the uncertainty of cluster membership, the distribution of each subject's highest class membership probability (the likelihood that reflects a subject's most likely cluster assignment) was evaluated (Supplementary Table 2). This evaluation was conducted by computing the median, and the lower and upper quartiles of these probabilities for all subjects within each cluster. The medians of the highest class membership probabilities ranged from 0.82 to 0.98 across clusters. These median values indicated a moderately high to very high degree of certainty in the assignment of subjects to their respective clusters.”

4. How are missing data handled?

Responses: Thank you for the question. In our data analyses, we did not encounter any missing data. However, it is important to acknowledge the potential presence of undiagnosed conditions or incomplete record-keeping within the electronic health record (EHR) database. These aspects require further attention and should be addressed in future studies.

The clustering variables were defined by the presence or absence of baseline medical conditions using the diagnosis codes in the EHR databases (ICD-9 codes in HK CDARS and Read codes in UK THIN). Therefore, every patient included in our cohorts had each condition either documented as present or considered absent. However, it is possible that some patients with the condition were not diagnosed or their conditions were not timely and accurately coded in the EHR database. We further discussed these issues in the limitations section of the discussion.

The following section is now added to the limitations section to further elaborate on the issue, with external validation proposed as the important next step:

[Discussion, page 16]

“...Moreover, underreporting of diagnoses due to factors such as underdiagnosis and undercoding is a common issue in EHR databases. The absence of a diagnosis code in the EHR databases, such as in undiagnosed patients, would result in the condition being classified as not present under our study's definition of diagnosis variables. For example, our reported baseline hypertensive disease prevalence of 13.1% in the UK THIN cohort was much lower than the prevalence of 29.7% in the general UK adult population reported by Health Survey for England (HSE).²⁵ This underestimation aligned with a study validating hypertension diagnosis coding in the UK THIN database, which reported an underestimated prevalence of 14.0% using Read codes to define hypertension.²⁵ The differences in the prevalence of clustering variables (such as hypertension) between HK CDARS and UK THIN could potentially contribute to the differences in the resulting clustering solutions. Therefore, a cautious interpretation of the clustering solutions is required. However, despite the variations between HK CDARS and UK THIN, our findings demonstrated consistency in several key aspects across the two cohorts, including the identification of subphenotypes with a pronounced presence of baseline CVE, and a temporal association between hip fracture subphenotypes and MACE. To reinforce the robustness of the LCA models, further independent validations across EHR databases and

populations are recommended, as consistent clustering results in diverse settings would support the generalisability of the LCA models. Future studies could also explore alternative definitions of conditions using data beyond diagnosis records.”

Reviewer #2 (Remarks to the Author):

• In this manuscript, the authors describe a machine learning protocol identifying clusters of comorbidities among individuals with hip fracture hospitalizations. The objective of the manuscript is to examine the association of so identified patient groups with the risk of adverse short-term cardiovascular events. Not surprisingly, the authors observed a positive association of pre-existing cardiovascular morbidities, such as heart failure, with major adverse cardiovascular event risk. The novelty of this finding depends very much on clinical practice. It would seem logical that the management of hip fracture patients would include, in addition to the fracture itself, also attendant comorbidities. Such information is readily available in clinical charts. The premise of this manuscript is therefore not very clear.

Responses: Thank you for providing us an opportunity to clarify the motivations of this study. Our study used LCA to integrate 22 baseline variables, creating comprehensive comorbidity profiles rather than examining individual conditions in isolation, as typically seen in clinical charts.

Without making any presumptions regarding which of the clustering variables would be particularly important, LCA highlighted two distinct subphenotypes with different enrichment in CVEs. Cluster 1 showed enrichment in cerebrovascular diseases and arterial disease (vascular diseases), while Cluster 2 exhibited enrichment in coronary heart disease, congestive heart failure, arrhythmia and conductive disorders (cardiac diseases).

Although the association between baseline CVE and increased post-fracture CVE risk may seem evident, and clinical guidelines acknowledge the consideration of multimorbidity, CVE risk management in hip fracture care is often under-emphasised in practice. The identified subphenotypes associated with higher risk of CVEs, all-cause mortality, and health care utilisation, when compared to the relatively healthy reference group, highlighted the critical need to recognise the heterogeneity among hip fracture patients.

The following section has now been added to the clinical implication section in the discussion for elaboration.

[Discussion, page 17-18]

“This study has important implications. Notably, the LCA results revealed subphenotypes with pronounced baseline CVE profiles across two independent population-based cohorts, without any preconceived assumption about CVE history being particularly important among the clustering variables. Although current clinical guidelines highlight the importance of perioperative evaluation and optimisation for patients with a high cardiac risk^{26 27}, there is a noticeable lack of emphasis on CVE risk management in the specific context of hip fracture care. For example, current heart failure management guidelines rarely cover the management of osteoporosis and hip fracture risk in heart failure patients^{28 29}. Our findings supported by the

previous studies emphasise the importance of taking measures to mitigate the risk of hip fracture among heart failure patients, given the potential poor prognosis after hip fracture. Notably, our temporal analysis showed an immediate increased risk of MACE after hip fracture, with the incidence rates of Cluster 2 in HK CDARS and Cluster 1 in UK THIN being particularly high. These findings underscore the importance of prompt CVE risk management especially in these high-risk patient groups.

The elevated CVE risk after hip fracture is inadequately documented in the literature, along with the association between pre-existing CVE and the subsequent risk of CVEs in hip fracture patients, especially considering multimorbidity is common in the older hip fracture patient population³⁰. Our study addressed this gap by quantifying the associated risks, and the elevated risk of MACE within 60 days post-hip fracture across all subphenotypes highlighted again the critical need for integrating CVE management into the broader hip fracture care strategy within the first year after hip fracture. Using the relatively healthy cluster as the reference group, this comparative framework allowed us to evaluate the relative prognosis of each unique comorbidity cluster, thereby laying the groundwork for more personalised management strategies and prioritisation of healthcare resources.

Hip fractures are often managed as a homogenous condition, yet the current study explored the heterogeneity among hip fracture patients and reported variations in mortality, CVE risk and hospital utilisation outcomes across subphenotypes. This variation emphasises the need to advance from conventional stratifications, such as those relying on a single baseline condition or demographic factor, to a more holistic, multimorbidity-focused approach in classifying patients, as facilitated by LCA. ...”

To further illustrate the hip fracture subphenotypes identified were not simple recapitulations of a traditional clinical group, we conducted an additional analysis (suggested by another reviewer) comparing the 60-day / 12-month event rates of MACE and mortality between the hip fracture subphenotypes and a reference cohort of MI patients identified in CDARS. This analysis allowed a comparison of the event rate gradients (from 60-day to 12-months) between the subphenotypes identified by LCA and a reference group of traditional CVD patients. Below is the new section added to address the observed differences:

[Results, page 10: paragraph 2]

“Furthermore, to verify the hip fracture subphenotypes were not direct replications of conventional cardiovascular disease (CVD) groups, a comparative analysis was conducted using a reference CDARS cohort composed of patients with myocardial infarction (MI). In this analysis, the 60-day and 12-month event rates for MACE and all-cause mortality for the subphenotypes were compared to that of the reference MI cohort (Supplementary Table 11). The MI cohort revealed expectedly high event rates of MACE and all-cause mortality. Notably, the MI cohort showed a moderate increase (28.48%) in the event rates of MACE (18.89% to 24.27%) and a substantial rise (137.8%) in mortality event rates (11.77% to 27.99%) from the 60-day to the 1-year period. Conversely, the hip fracture cohort, with 3.18% of patients with

baseline MI, demonstrated a sharper increase in both outcomes over the same period, with the MACE event rates more than doubling (164.2% increase, from 2.29% to 6.05%) and all-cause mortality rates tripling (255.2% increase, from 4.67% to 16.59%). This trend was consistent across all three clusters within the hip fracture cohort. In particular, Cluster 2 showed the highest absolute event rates among the hip fracture clusters, with the 1-year MACE rate being 10.72%, almost half of that of the MI cohort, even though only 15.04% of patients in Cluster 2 had baseline MI. In addition, the 1-year mortality rate in Cluster 2 surpassed that of the MI cohort (31.35% vs 27.99%).”

Supplementary Table 11. Comparison of the event rates of the hip fracture subphenotypes with the MI reference cohort in HK CDARS.

	Cluster 1	Cluster 2	Cluster 3	Total	MI Reference Cohort
N	16,762	9,860	51,795	78,417	36,933
Baseline MI	3.42%	15.04%	0.85%	3.18%	100%
MACE*					
60-days	3.48%	4.11%	1.56%	2.29%	18.89%
1-year	8.33%	10.72%	4.42%	6.05%	24.27%
All-cause mortality					
60-days	4.53%	9.59%	3.78%	4.67%	11.77%
1-year	17.25%	31.35%	13.57%	16.59%	27.99%

Notes: MACE = Major Adverse Cardiovascular; MI = Myocardial infarction.

*MACE defined by MI and stroke (as heart failure hospitalisation data were not available in the MI Reference Cohort).

[Discussion, page 13, paragraph 2]

“...Such findings could be driven by the higher prevalence of prevalent cardiac events in these clusters and other potential unmeasured confounders. However, the differences in the event rates of MACE and all-cause mortality between the hip fracture subphenotypes and the reference group of MI patients partially supported that the subphenotypes identified by LCA were not simply reiterations of traditional CVD patient groups. ...”

• The authors do not include information on the severity of the hip fractures or anything related to hip replacement. Further, analyses are not stratified by age categories or sex, which would provide additional information pertaining to likelihood of cardiovascular outcomes.

Responses: Thank you for the comments. Systematic records on hip fracture severity are not directly available in the EHR databases of this study. However, Hong Kong CDARS hospital database contains records of hip surgeries, and the surgery types could serve as a surrogate marker of hip fracture severity. Following your suggestion, we examined the most common procedures associated with hip fracture events among patients in the

HK CDARS cohort. We identified that internal fixation (n = 24,526) and partial hip replacement (n = 14,042) were the two most commonly recorded procedures. As a result, we now added a subgroup analysis evaluating CVE risk in patients in these two subgroups (Supplementary Table 8). Furthermore, we agree that stratification of age categories and sex is important for a more comprehensive investigation of CVE risk. In response to your suggestion, we have included the subgroup analyses by age categories and sex (Supplementary Table 8).

The following sections on the stratified analysis have been added to the text:

[Results, page 8, paragraph 2; page 9, paragraph 1]

“...The increased risk of MACE and all-cause mortality reported in the main analysis in Cluster 1 and 2 in HK CDARS, and Cluster 1 in UK THIN, were in general consistently observed across all the age, sex and surgery type subgroups (Supplementary Table 8). However, differences in the magnitude of these risks were observed when comparing the different subgroups. The stratified analysis generally showed that females, younger patients, and those undergoing partial hip replacement surgeries tended to exhibit higher risks, as reflected by HRs, when compared with the male, older, and internal fixation counterparts. In particular, in Cluster 2 in HK CDARS and Cluster 1 in UK THIN, the HR for the association between the hip fracture subphenotype and MACE was observed to be higher in females when compared to males (interaction p-value <0.05).”

[Discussion, page 12: paragraph 2, last sentence; page 13, paragraph 1]

“...A sex-specific risk of MACE has been observed in Cluster 2 in HK CDARS and Cluster 1 in UK THIN. While male is often known to be associated with a higher risk of post-fracture morbidity including CVE,²¹ our subgroup analysis showed that in Cluster 2 in HK CDARS and Cluster 1 in UK THIN, the HR for the association between the hip fracture subphenotype and MACE was higher in females than in males. Therefore, it is important to further consider the factor of sex when evaluating the prognosis for those within the already high-risk subphenotypes.”

[Method, page 25, paragraph 1]

“...Furthermore, stratified analyses were conducted to evaluate the associations between the hip fracture subphenotypes and the outcomes of MACE and mortality, across sex, age, and the types of surgical treatment received (internal fixation and partial hip replacement).”

• The manuscript includes language that is suggestive of a causality between hip fractures and cardiovascular outcomes. This is likely not intentional, however, whereas it is appropriate to discuss the association of comorbidity clusters with cardiovascular outcomes among hip fracture patients, it is not appropriate to make a statement about the association of hip fractures with cardiovascular outcomes (e.g. page 8, line 169), as the causal mechanism is not there. This is further incorrect, given that the study population included exclusively patients with hip fractures.

Responses: Thank you for your feedback on the language and this is indeed not

intentional. We have revised the language in the manuscript to avoid suggesting any causal link between hip fractures and cardiovascular outcomes. We also ensure the terms “clusters/subphenotypes” and “hip fractures” are clearly differentiated.

• I had a difficult time understanding the findings presented in Figure 3. It would appear that each of the identified clusters was associated with MACE at 60 days post-admission. The maintenance of this risk within an extended 180-day window does not imply a dose relationship, as the figure suggests. Partitioning the risk into time periods, as presented in Figure 3, necessitates a re-adjustment of the denominators, which does not appear to have been done. The relevance of the identified clusters in terms of MACE risk is not clear from this figure.

Responses: Thank you for the comments. Figure 3 (now Figure 2) shows the results of the SCCS. We further elucidated the method and added the motivation for doing this analysis after clustering earlier in the manuscript.

The details of the SCCS were described previously(Petersen I, Douglas I, Whitaker H. Self controlled case series methods: an alternative to standard epidemiological study designs BMJ 2016;354 :i4515). In brief, the analysis focused only on hip fracture patients who experienced the event. There were distinct risk periods following exposure (hip fracture), and time periods when the patient was not at risk were used as a reference. Therefore, each patient served as their own control, effectively accounting for time-fixed confounders and minimising their impact on the analysis. No dose relationship can be implied by SCCS. We used the method to investigate the temporal association patterns in each LCA-derived cluster.

[Introduction, page 5, paragraph 1]

“...Unlike the conventional survival analyses comparing the prognosis risk between individuals, SCCS analyses focused only on hip fracture patients who experienced the prognostic events with each patient serving as their own control, which effectively controlled for time-fixed confounders and between-individual differences¹³. Consequently, SCCS analyses enabled the detection of temporal patterns within each LCA-derived cluster and facilitated the identification of potential differences in temporal patterns across the clusters, which cannot be captured by conventional survival analyses alone.”

And in the discussion:

[Results, page 9, paragraph 3]

“...Similar temporal association patterns were observed in all clusters, including the relatively healthy cluster. Specifically, the age-adjusted incidence rate ratios (IRRs) of MACE were found to be the highest and statistically significant at 1-60 days after hip fracture across the three clusters (Cluster 1: IRR 1.84, 95% CI 1.54 to 2.20; Cluster 2: IRR 1.99, 95% CI 1.62 to 2.45; Cluster 3: IRR 2.05, 95% CI 1.81 to 2.33), and the IRRs decreased during the subsequent post-hip fracture risk periods (Fig. 2). The cluster-specific patterns matched the overall temporal pattern of the entire cohort, with the age-adjusted IRR of MACE at 1-60 days after hip fracture

being 1.93 (95% CI, 1.76 to 2.11; Fig. 2). Similar results were observed in the UK THIN database, where significantly higher IRRs of MACE were observed at 1-60 days after hip fracture in the two clusters (Cluster 1: IRR 3.35, 95% CI 2.28 to 4.92; Cluster 2: IRR 2.25, 95% CI 1.81 to 2.81), matching the overall pattern of the entire cohort (IRR 2.43, 95% CI 2.01 to 2.94; Fig. 2)."

[Results, page 10, paragraph 1]

"...In addition, we plotted the incidence rates of MACE within one year following hip fracture (Supplementary Fig. 6). The plots showed similar temporal patterns across clusters, but there were notable differences in the incidence rates across clusters in both HK CDARS and UK THIN, which aligned with the results from competing risk regression and SCCS."

[Discussion, page 13, paragraph 2]

"...We further conducted SCCS analyses to address the between-person residual confounding and to investigate the temporal patterns across clusters, which cannot be achieved through conventional survival analyses. While the results revealed similar temporal association patterns across clusters, consistently showing the highest risk of MACE within 1-60 days after the hip fracture, it is important to note that the risk of post-hip fracture MACE remained elevated even among individuals classified as relatively healthy. ..."

And regarding the re-adjustment of the denominators, we partitioned the risk periods into shorter time intervals (30 days rather than 60 days) and considered a different reference time period (only the post-hip fracture period as the baseline; Supplementary Table 10).

Reviewer #3 (Remarks to the Author):

The manuscript by Hsu and colleagues entitled “Unveiling Unique Clinical Phenotypes of Hip Fracture Patients and the Temporal Association with Cardiovascular Events in Hong Kong and the United Kingdom: A Retrospective Study” describes findings from two prospective cohort studies of people ≥ 65 years with hip fractures, the HK CDARS hip fracture cohort (n=78,417) and the UK THIN cohort (n=27,968) investigating the use of latent class analysis to identify clusters or sub phenotypes of hip fracture patients based and investigate their association with incident cardiovascular events. The authors found that there were three sub phenotypes (clusters) in the HK CDARS and two sub phenotypes in the UK THIN cohorts. The sub phenotypes with atherosclerotic disease (cluster 1) and that with heart failure (cluster 2) were at 2-4 times higher relative hazard for cardiovascular events. With this risk being most evident in 1-60 days post hip fracture. This is noteworthy as post-hip fracture risk of CVD events is not well described in literature or recognized more widely as a CVD risk factor. The manuscript is very well written, is original and provides enough details to be reproduced.

Strengths are that the study has utilised two large hip fracture cohorts and have carefully designed and undertaken this work. The results and findings are of great interest and potentially clinically meaningful.

Limitations of the study are that the results section is quite difficult to follow, and the discussion is quite cursory in parts i.e., does not discuss how these findings may be operationalised in a clinical setting such as hospitals or fracture liaison services and does not discuss how hip fracture is a non-traditional gender-specific cardiovascular disease risk factor and how the findings from this manuscript may contribute to ongoing efforts to bridge improve CVD outcomes in women.

Responses:

Thank you for your comments. We have now included a concise paragraph in the introduction to summarise our workflow and provide an overview of the rationale behind each method used. Furthermore, each results section now starts with one sentence illustrating the method’s purpose. These additions aim to assist readers in comprehending our results section more effectively and following the logical progression of our study.

[Introduction, page 4, paragraph 3; page 5, paragraph 1]

“Clustering algorithms are commonly used to classify patients with similar characteristics or features. Among various clustering algorithms, latent class analysis (LCA), an unsupervised machine learning technique for classifying subjects into clusters by a combination of variables⁹⁻¹¹, has been shown to be the optimum clustering algorithm for health records¹². Therefore, in this study, we first utilised the unsupervised LCA to subgroup hip fracture patients who share common clinical characteristics in two extensive hip fracture cohorts: one from Hong Kong (HK; N=78,417) and the other from the United Kingdom (UK; N=27,948). Subsequently, to uncover variations in long-term outcomes for hip fracture patients based on their subphenotypic classification, conventional survival analyses (between-individual comparison) and self-

controlled case series (SCCS; within-individual comparison) were conducted. The analyses aimed to quantify the risk of CVE-related outcomes, all-cause mortality, and health service utilisation following hip fracture and to evaluate potential differences across the identified subphenotypes. Unlike the conventional survival analyses comparing the prognosis risk between individuals, SCCS analyses focused only on hip fracture patients who experienced the prognostic events with each patient serving as their own control, which effectively controlled for time-fixed confounders and between-individual differences¹³. Consequently, SCCS analyses enabled the detection of temporal patterns within each LCA-derived cluster and facilitated the identification of potential differences in temporal patterns across the clusters, which cannot be captured by conventional survival analyses alone. Overall, the conventional survival analysis and SCCS are complementary approaches that provide a comprehensive understanding of the prognosis risk and temporal associations in the identified subphenotypes.”

Regarding the clinical implications of this study, we revised our discussion to better connect the findings from our study with clinical management (new sections are provided below). CVE risk management in hip fracture care is often under-emphasised in practice. We showed some LCA-derived clusters having higher incidence rates and risk of post-hip fracture MACE, highlighting the critical need to recognise the heterogeneity among hip fracture patients. To facilitate the operationalisation of our findings, we have included a discussion on the potential development of a digital tool in the future, which will allow researchers and clinicians to construct hip fracture subphenotypes using their EHR databases. Such a tool could potentially encourage the development of population-specific subphenotyping models for hip fracture patients.

[Discussion, page 17-18]

“This study has important implications. Notably, the LCA results revealed subphenotypes with pronounced baseline CVE profiles across two independent population-based cohorts, without any preconceived assumption about CVE history being particularly important among the clustering variables. Although current clinical guidelines highlight the importance of perioperative evaluation and optimisation for patients with a high cardiac risk^{24 25}, there is a noticeable lack of emphasis on CVE risk management in the specific context of hip fracture care. For example, current heart failure management guidelines rarely cover the management of osteoporosis and hip fracture risk in heart failure patients^{28 29}. Our findings supported by the previous studies emphasise the importance of taking measures to mitigate the risk of hip fracture among heart failure patients, given the potential poor prognosis after hip fracture. Notably, our temporal analysis showed an immediate increased risk of MACE after hip fracture, with the incidence rates of Cluster 2 in HK CDARS and Cluster 1 in UK THIN being particularly high. These findings underscore the importance of prompt CVE risk management especially in these high-risk patient groups.

The elevated CVE risk after hip fracture is inadequately documented in the literature, along with the association between pre-existing CVE and the subsequent risk of CVEs in hip fracture patients, especially considering multimorbidity is common in the older hip fracture patient population.³⁰ Our study addressed this gap by quantifying the associated risks, and the elevated risk of MACE within 60 days post-hip fracture across all subphenotypes highlighted again the critical need for integrating CVE management into the broader hip fracture care strategy within

the first year after hip fracture. Using the relatively healthy cluster as the reference group, this comparative framework allowed us to evaluate the relative prognosis of each unique comorbidity cluster, thereby laying the groundwork for more personalised management strategies and prioritisation of healthcare resources.

Hip fractures are often managed as a homogenous condition, yet the current study explored the heterogeneity among hip fracture patients and reported variations in mortality, CVE risk and hospital utilisation outcomes across subphenotypes. This variation emphasises the need to advance from conventional stratifications, such as those relying on a single baseline condition or demographic factor, to a more holistic, multimorbidity-focused approach in classifying patients, as facilitated by LCA.”

[Discussion, page 18, paragraph 2]

“...To operationalise the subphenotyping process, a digital tool could be developed and incorporated into the electronic clinical management system for personalised management of the hip fracture patients based on their subphenotypes. It is crucial to acknowledge the population-specific nature of EHR databases and the resulting LCA models. Thus, population-specific subphenotyping models should be developed, instead of one single model applying to all populations. Future studies should also be conducted to evaluate if personalised treatment based on LCA-derived subphenotypes is clinically useful in reducing the risk of CVE in patients with hip fracture.”

Furthermore, we agree on the importance of discussing hip fracture as a non-traditional sex-specific risk factor of CVE. We have now added an additional survival analysis stratified by sex, providing a quantification of the sex-specific risk between the hip fracture subphenotypes (Supplementary Table 8). In particular, our results indicated that while patients in Cluster 2 in HK CDARS and Cluster 1 in UK THIN already exhibited an elevated risk of MACE and mortality, the risk was further escalated in female patients. The findings highlighted the need for targeted management of CVE risk for female patients belonging to these high-risk subphenotypes. Below are the relevant sections newly added to the manuscript:

[Results, page 8, paragraph 2]

“...The increased risk of MACE and all-cause mortality reported in the main analysis in Cluster 1 and 2 in HK CDARS, and Cluster 1 in UK THIN, were in general consistently observed across all the age, sex and surgery type subgroups (Supplementary Table 8). However, differences in the magnitude of these risks were observed when comparing the different subgroups. The stratified analysis generally showed that females, younger patients, and those undergoing partial hip replacement surgeries tended to exhibit higher risks, as reflected by HRs, when compared with the male, older, and internal fixation counterparts. In particular, in Cluster 2 in HK CDARS and Cluster 1 in UK THIN, the HR for the association between the hip fracture subphenotype and MACE was observed to be higher in females when compared to males (interaction p-value <0.05).”

[Discussion, page 12: paragraph 2, last sentence; page 13: paragraph 1]

“...A sex-specific risk of MACE has been observed in Cluster 2 in HK CDARS and Cluster 1 in UK THIN. While male is often known to be associated with a higher risk of post-fracture morbidity including CVE²¹, our subgroup analysis showed that in Cluster 2 in HK CDARS and Cluster 1 in UK THIN, the HR for the association between the hip fracture subphenotype and MACE was higher in females than in males. Therefore, it is important to further consider the factor of sex when evaluating the prognosis for those within the already high-risk subphenotypes.”

Discussion on how the hip-fracture 60 day and 12-month MACE events rates compare to the post-myocardial infarction 60 day and 12-month MACE and mortality events rates would be informative for the readers to allow comparison for the observed gradient of risk between these clusters.

Responses: Thank you for your valuable suggestion. We have incorporated an analysis comparing the 60-day / 12-month event rates between the hip fracture cohort and an MI cohort we identified in CDARS. The results are shown in Supplementary Table 11.

[Results, page 10, paragraph 2]

“Furthermore, to verify the hip fracture subphenotypes were not direct replications of conventional cardiovascular disease (CVD) groups, a comparative analysis was conducted using a reference CDARS cohort composed of patients with myocardial infarction (MI). In this analysis, the 60-day and 12-month event rates for MACE and all-cause mortality for the subphenotypes were compared to that of the reference MI cohort (Supplementary Table 11). The MI cohort revealed expectedly high event rates of MACE and all-cause mortality. Notably, the MI cohort showed a moderate increase (28.48%) in the event rates of MACE (18.89% to 24.27%) and a substantial rise (137.8%) in mortality event rates (11.77% to 27.99%) from the 60-day to the 1-year period. Conversely, the hip fracture cohort, with 3.18% of patients with baseline MI, demonstrated a sharper increase in both outcomes over the same period, with the MACE event rates more than doubling (164.2% increase, from 2.29% to 6.05%) and all-cause mortality rates tripling (255.2% increase, from 4.67% to 16.59%). This trend was consistent across all three clusters within the hip fracture cohort. In particular, Cluster 2 showed the highest absolute event rates among the hip fracture clusters, with the 1-year MACE rate being 10.72%, almost half of that of the MI cohort, even though only 15.04% of patients in Cluster 2 had baseline MI. In addition, the 1-year mortality rate in Cluster 2 surpassed that of the MI cohort (31.35% vs 27.99%).”

Supplementary Table 11. Comparison of the event rates of the hip fracture subphenotypes with the MI reference cohort in HK CDARS.

	Cluster 1	Cluster 2	Cluster 3	Total	MI Reference Cohort
N	16,762	9,860	51,795	78,417	36,933
Baseline MI	3.42%	15.04%	0.85%	3.18%	100%
MACE*					
60-days	3.48%	4.11%	1.56%	2.29%	18.89%

1-year	8.33%	10.72%	4.42%	6.05%	24.27%
All-cause mortality					
60-days	4.53%	9.59%	3.78%	4.67%	11.77%
1-year	17.25%	31.35%	13.57%	16.59%	27.99%

Notes: MACE = Major Adverse Cardiovascular; MI = Myocardial infarction.

*MACE defined by MI and stroke (as heart failure hospitalisation data were not available in the MI Reference Cohort).

[Discussion, page 13, paragraph 2]

“...Such findings could be driven by the higher prevalence of prevalent cardiac events in these clusters and other potential unmeasured confounders. However, the differences in the event rates of MACE and all-cause mortality between the hip fracture subphenotypes and the reference group of MI patients partially supported that the subphenotypes identified by LCA were not simply reiterations of traditional CVD patient groups. ...”

The cohorts' clinical characteristics and particularly heart failure, hypertensive disease, diabetes, osteoporosis and renal diseases are very different between the cohorts, and the clustering solutions were similarly different. This makes interpretation and operationalisation of the findings difficult. Greater discussion of this issue and the next steps needed are warranted.

Responses:

Thank you for your valuable insights. We have elaborated on the differences between the two cohorts and outlined proposed next steps accordingly. In particular, we emphasised the importance of external validation as a critical next phase to facilitate the adoption of our LCA models.

[Discussion, page 15: last 2 sentences; page 16-17]

“Additionally, the clustering solutions generated by LCA were not identical in the two cohorts. The HK CDARS and UK THIN databases differed significantly not only in terms of clinical settings (hospital-based in HK CDARS versus primary care in UK THIN), but also in the demographic composition of their populations (predominantly Asian Chinese in HK CDARS versus predominantly Caucasian in UK THIN), data entry behaviours, and diagnosis coding systems. These diverse factors inherent to EHR databases could affect the generalisability of LCA models and survival analysis results. These variations may also explain the presence of an additional CVE cluster in the Hong Kong cohort, potentially attributed to a more comprehensive and timely capture of severe conditions in the hospital-based CDARS when compared to the primary care-based THIN. Moreover, underreporting of diagnoses due to factors such as underdiagnosis and undercoding is a common issue in EHR databases. The absence of a diagnosis code in the EHR databases, such as in undiagnosed patients, would result in the condition being classified as not present under our study's definition of diagnosis variables. For example, our reported baseline hypertensive disease prevalence of 13.1% in the UK THIN cohort was much lower than the prevalence of 29.7% in the general UK adult population reported by Health Survey for England (HSE)²⁵. This underestimation aligned with a study validating hypertension diagnosis coding in the UK THIN database, which reported an underestimated prevalence of 14.0% using Read codes to define hypertension²⁵. The

differences in the prevalence of clustering variables (such as hypertension) between HK CDARS and UK THIN could potentially contribute to the differences in the resulting clustering solutions. Therefore, a cautious interpretation of the clustering solutions is required. However, despite the variations between HK CDARS and UK THIN, our findings demonstrated consistency in several key aspects across the two cohorts, including the identification of subphenotypes with a pronounced presence of baseline CVE, and a temporal association between hip fracture subphenotypes and MACE. To reinforce the robustness of the LCA models, further independent validations across EHR databases and populations are recommended, as consistent clustering results in diverse settings would support the generalisability of the LCA models. Future studies could also explore alternative definitions of conditions using data beyond diagnosis records.”

[Discussion, page 18, paragraph 2]

“...To operationalise the subphenotyping process, a digital tool could be developed and incorporated into the electronic clinical management system for personalised management of the hip fracture patients based on their subphenotypes. It is crucial to acknowledge the population-specific nature of EHR databases and the resulting LCA models. Thus, population-specific subphenotyping models should be developed, instead of one single model applying to all populations. Future studies should also be conducted to evaluate if personalised treatment based on LCA-derived subphenotypes is clinically useful in reducing the risk of CVE in patients with hip fracture.”

The composite components for the MACE outcome are not commonly used (Bosco, Elliott, et al. "Major adverse cardiovascular event definitions used in observational analysis of administrative databases: a systematic review." BMC Medical Research Methodology 21.1 (2021): 1-18). In the HK CDARS the increase in the Heart failure events did appear substantially greater than for the other components of MACE. This outcome should be justified by a reference or previous work. It would also be good to compare to a more commonly used definitions such as AMI, stroke and all-cause mortality.

Responses: Thank you for your comments. We added a reference that previously used the same MACE definition as ours to justify our choice [1]. We acknowledge that it is important to compare the current definition in HK CDARS with a more widely adopted definition. To address this, we conducted a sensitivity analysis using MACE defined by AMI+stroke only, which was identified as the most common definition in the systematic review by Bosco, Elliott, et al.[2] The conclusions of the sensitivity analysis results were essentially unchanged (Supplementary Table 7).

[Method, page 25, paragraph 1]

“...Another sensitivity analysis was also performed using a more conventional definition of MACE in HK CDARS, which included only acute myocardial infarction and stroke [2].”

[1] Arnaout R, Nah G, Marcus G, Tseng Z, Foster E, Harris IS, Divanji P, Klein L, Gonzalez J, Parikh N. Pregnancy complications and premature cardiovascular events among 1.6 million California pregnancies. *Open Heart*. 2019 Feb 27;6(1):e000927. doi: 10.1136/openhrt-2018-000927. PMID: 30997125; PMCID: PMC6443129.

[2] Bosco E, Hsueh L, McConeghy KW, Gravenstein S, Saade E. Major adverse cardiovascular event definitions used in observational analysis of administrative databases: a systematic review. *BMC Med Res Methodol.* 2021 Nov 6;21(1):241. doi: 10.1186/s12874-021-01440-5. PMID: 34742250; PMCID: PMC8571870.

Minor

Figure 3 - appears redundant give Table 1.

Responses: Thank you for your comments. Table 1 summarised the baseline characteristics of the three hip fracture subphenotypes identified by LCA using the CDARS data, without evaluating any clinical outcomes of interest. In contrast, the old Figure 3 showed the results of SCCS investigating the temporal association of the three hip fracture subphenotypes with the outcome of MACE. Figure 3 and Table 1 present distinct aspects of our findings.

Upon further inspection, the old Figure 1 may appear redundant given Table 1, since Figure 1 is simply a visual representation of the proportions presented in Table 1. We have now moved Figure 1 to the supplement (the new Supplementary Figure 3).

Abstract: Some quantification of the “higher risks of major adverse cardiovascular events” should be included in the abstract.

Responses: Thank you for bringing this to our attention. Quantification of MACE risk is now added to the abstract, with the revised statement shown here:

[Abstract, page.3]

“Compared to Cluster 3, higher risks of major adverse cardiovascular events (MACE) were observed in Cluster 1 (**hazard ratio [HR] 1.97, 95% CI 1.83 to 2.12**) and Cluster 2 (**HR 4.06, 95% CI 3.78 to 4.35**). Clusters 1 and 2 were also associated with higher risk of mortality, more unplanned accident and emergency (A&E) visits, and longer hospital stays.”

Abstract: The study abstract aims sentence “This study aimed to identify subphenotypes of hip fracture patients and investigate their association with incident CVEs in Hong Kong (HK) and the United Kingdom (UK) populations.” fails to mention all-cause mortality and health service utilisation as outcomes, despite reporting the results of these outcomes.

Responses: Thank you for the comments. We added all-cause mortality and health service utilisation outcomes in the aim statement of the abstract. Below is the revised statement:

[Abstract, page 3]

“This study aimed to identify subphenotypes of hip fracture patients and investigate their association with incident CVEs, **all-cause mortality**, and **health service utilisation** in Hong Kong (HK) and the United Kingdom (UK) populations.”

REVIEWERS' COMMENTS

Reviewer #1 (Remarks to the Author):

The authors did a thorough job of addressing the comments that I provided in my first review of the paper as well as the concerns of other reviewers. I believe that their responses and edits to my questions are more than satisfactory (as I summarize below), and I have no further substantive remarks to offer. The motivation for their analysis is clear, their statistical methods are valid and well-supported, and their discussions and arguments across various sections are compelling (and substantially more robust than before). If anything, the Discussion section has some repeated ideas relative to strengths/aims of the paper. For example, I think this last bit could be removed (but it's up to the authors' discretion):

"In conclusion, this study identified distinct subphenotypes of hip fracture in both the Hong Kong and UK older adult populations using LCA. Temporal associations with MACE were observed in all hip fracture patient clusters. Notably, heart failure consistently emerged as a key characteristic associated with poor prognosis in hip fracture patients. Personalised care of hip fracture patients, considering their specific subphenotypes, is required to prevent MACE."

Authors' response to major comments: The authors remediated two major limiting elements in their exposition of methods. First, other reviewers and I noted that the clustering of hip fracture patients into sub-phenotypes based on measures of cardiovascular health might create already known high risk groups for MACE in such a way that the corresponding survival analysis appears self-evident. They not only clarify the purpose and value of clustering based on these variables, but they also illustrate that the clusters are not simply copies of traditional clinical groups (using a reference MI patient cohort). Second, I was concerned about how the SCCS analysis complemented clustering and conventional survival analyses, and they sufficiently supported its purpose and re-articulated the accompanying results.

Authors' response to minor comments: They further elaborated on how a single Pearson's correlation coefficient is obtained for cluster comparison. They detailed their rationale for selecting the optimal number of clusters in LCA. They included a table in the supplemental materials to assess cluster membership uncertainty and added a statement about it in the text. They provided a discussion of the potential impact of missing/misreported data coming from EHR databases.

Reviewer #2 (Remarks to the Author):

The authors' response to my previous comments and concerns is sufficient. A minor additional stylistic comment: Line 258, page 11: Use the term "male sex" instead of "male" at the beginning of the sentence.

Reviewer #3 (Remarks to the Author):

The authors have addressed all of my concerns.

REVIEWERS' COMMENTS

Reviewer #1 (Remarks to the Author):

The authors did a thorough job of addressing the comments that I provided in my first review of the paper as well as the concerns of other reviewers. I believe that their responses and edits to my questions are more than satisfactory (as I summarize below), and I have no further substantive remarks to offer. The motivation for their analysis is clear, their statistical methods are valid and well-supported, and their discussions and arguments across various sections are compelling (and substantially more robust than before). If anything, the Discussion section has some repeated ideas relative to strengths/aims of the paper. For example, I think this last bit could be removed (but it's up to the authors' discretion):

"In conclusion, this study identified distinct subphenotypes of hip fracture in both the Hong Kong and UK older adult populations using LCA. Temporal associations with MACE were observed in all hip fracture patient clusters. Notably, heart failure consistently emerged as a key characteristic associated with poor prognosis in hip fracture patients. Personalised care of hip fracture patients, considering their specific subphenotypes, is required to prevent MACE."

Authors' response to major comments: The authors remediated two major limiting elements in their exposition of methods. First, other reviewers and I noted that the clustering of hip fracture patients into sub-phenotypes based on measures of cardiovascular health might create already known high risk groups for MACE in such a way that the corresponding survival analysis appears self-evident. They not only clarify the purpose and value of clustering based on these variables, but they also illustrate that the clusters are not simply copies of traditional clinical groups (using a reference MI patient cohort). Second, I was concerned about how the SCCS analysis complemented clustering and conventional survival analyses, and they sufficiently supported its purpose and re-articulated the accompanying results.

Authors' response to minor comments: They further elaborated on how a single Pearson's correlation coefficient is obtained for cluster comparison. They detailed their rationale for selecting the optimal number of clusters in LCA. They included a table in the supplemental materials to assess cluster membership uncertainty and added a statement about it in the text. They provided a discussion of the potential impact of missing/misreported data coming from EHR databases.

Response: Thank you again for your time on the review and for providing us the opportunity to improve the study before.

We agree that the final paragraph of the Discussion contained repeated ideas mentioned before, and we have now removed the last paragraph.

Reviewer #2 (Remarks to the Author):

The authors' response to my previous comments and concerns is sufficient. A minor additional stylistic comment: Line 258, page 11: Use the term "male sex" instead of "male" at the beginning of the sentence.

Response: Thank you again for your time on the review and the valuable feedback on the study. The comment on sex has now been addressed:

[Discussion, page 12, paragraph 1]

“While the male sex is often known to be associated with a higher risk of post-fracture morbidity including CVE²¹, our subgroup analysis showed that in Cluster 2 in HK CDARS and Cluster 1 in UK THIN, the HR for the association between the hip fracture subphenotype and MACE was higher in females than in males.”

Reviewer #3 (Remarks to the Author):

The authors have addressed all of my concerns.

Response: Thank you again for your time on the review and the valuable comments on the study.